# N6-methyladenosine RNA modification promotes Severe Fever with Thrombocytopenia Syndrome Virus infection

Zhiqiang Chen[1,2☉], Jinyu Zhang[1☉], Jun Wang[1☉], Hao Tong[1], Wen Pan[1], Feng Ma[3,4]\*, Qihan Wu[5]\*, Jianfeng Dai[1]\*

1 Jiangsu Key Laboratory of Infection and Immunity, MOE Key Laboratory of Geriatric Diseases and Immunology, The Forth Affiliated Hospital of Soochow University, Institutes of Biology and Medical Sciences, Suzhou Medical College of Soochow University, Suzhou, China, 2 Department of Nuclear Medicine, The First Affiliated Hospital of Soochow University, Suzhou, China, 3 CAMS Key Laboratory of Synthetic Biology Regulatory Elements, Institute of Systems Medicine, Chinese Academy of Medical Sciences & Peking Union Medical College, Beijing, China, 4 Suzhou Institute of Systems Medicine, Suzhou, China, 5 Shanghai-MOST Key Laboratory of Health and Disease Genomics, NHC Key Laboratory of Reproduction Regulation, Shanghai Institute for Biomedical and Pharmaceutical Technologies, Shanghai, China

☉ These authors contributed equally to this work.
\* maf@ism.pumc.edu.cn (FM); wuqihan@sibpt.com (QW); daijianfeng@suda.edu.cn (JD)

**Data Availability Statement:** All sequencing data in this study have been deposited at the National Genomics Data Center (https://ngdc.cncb.ac.cn/)

## Abstract

Severe Fever with Thrombocytopenia Syndrome Virus (SFTSV), a novel bunyavirus primarily transmitted by *Haemaphysalis longicornis*, induces severe disease with a high mortality rate. N6-methyladenosine ($m^6A$) is a prevalent internal chemical modification in eukaryotic mRNA that has been reported to regulate viral infection. However, the role of $m^6A$ modification during SFTSV infection remains elusive. We here reported that SFTSV RNAs bear $m^6A$ modification during infection. Manipulating the expressions or activities of host $m^6A$ regulators significantly impacted SFTSV infection. Mechanistically, SFTSV recruited $m^6A$ regulators through the nucleoprotein to modulate the $m^6A$ modification of viral RNA, eventually resulting in enhanced infection by promoting viral mRNA translation efficiency and/or genome RNA stability. $m^6A$ mutations in the S genome diminished virus particle production, while $m^6A$ mutations in the G transcript impaired the replication of recombinant vesicular stomatitis virus (rVSV) expressing G protein *in vitro* and *in vivo*. Interestingly, $m^6A$ modification was evolutionarily conserved and facilitated SFTSV infection in primary tick cells. These findings may open an avenue for the development of $m^6A$-targeted anti-SFTSV vaccines, drugs, and innovative strategies for the prevention and control of tick-borne disease.

## Author summary

SFTSV, or Dabie bandavirus (DBV), is a tick-transmitted bunyavirus that causes severe fever with thrombocytopenia syndrome with a mortality rate as high as 12%. *Haemaphysalis longicornis* is the main transmission vector of SFTSV. The absence of therapy and vaccines against SFTSV infection poses an urgent need to better understand the infection mechanism of SFTSV to develop effective antiviral drugs or vaccines. $m^6A$ is one of the

under the accession number GSA: CRA019122.
https://ngdc.cncb.ac.cn/gsa/browse/CRA019122.

**Funding:** This research was supported by grants from the Priority Academic Program Development of Jiangsu Higher Education Institutions (PAPD), National Natural Science Foundation of China (32170142 and 81971917 to J.D.), Jiangsu Natural Science Foundation (BK20211310 to J.D.), CAMS Initiation Fund for Medical Sciences (CIFMS, 2023-I2M-2-010 and 2022-I2M-2-004 to F.M.), the Suzhou Municipal Key Laboratory (SZS2023005 to F.M.), and fundings from the NCTIB Fund for R&D Platform for Cell and Gene Therapy(F.M.), the 333 High-level Talent Training Project (F.M.), and Suzhou International Joint Laboratory for Diagnosis and Treatment of Brain Diseases (J.D.). The funders had no role in study design, data collection and analysis, decision to publish, or preparation of the manuscript.

**Competing interests:** The authors have declared that no competing interests exist.

most prevalent internal modifications in mRNAs in most eukaryotes and is involved in the modulation of viral infection. However, the biological functions of RNA m$^6$A modification during virus-host or virus-vector interactions are still controversial. Our results show that SFTSV RNAs bear m$^6$A modification which is utilized by SFTSV to promote viral infection. SFTSV recruits m$^6$A regulators through the nucleoprotein to modulate the m$^6$A modification of viral RNA and eventually results in enhanced infection by influencing the translation efficiency of viral mRNA and the stability of genome RNA. Interestingly, RNA m$^6$A modification is evolutionarily conserved and facilitates SFTSV infection in primary tick cells. Thus, targeting m$^6$A may serve as a novel strategy to develop tailored anti-SFTSV pharmaceuticals, preventive vaccines, and management of tick-borne diseases.

## Introduction

Severe Fever with Thrombocytopenia Syndrome (SFTS) denotes a tick-borne infectious disease rooted in Severe Fever with Thrombocytopenia Syndrome Virus (SFTSV) infection [1]. Since its initial emergence in 2009, the geographic spread of SFTS has steadily burgeoned. Beyond China's 23 provinces [2], nations like South Korea, Japan, and Vietnam have sequentially reported related cases [3–5]. The mortality rate among SFTS patients reaches 12% [2,4,6]. The long-horned tick *Haemaphysalis longicornis* (*H. longicornis*) serves as a primary vector for SFTSV transmission. SFTS has emerged as a foremost global infectious disease [7], presenting a significant menace to public health. Urgent demand exists for the development of novel and specific therapeutic drugs or preventive vaccines targeting SFTSV.

SFTSV, categorized as a segmented negative-strand RNA virus (sNSV), manifests its genome across three segments denoted as L, M, and S [1]. The L segment spans a length of 6368 nucleotides (nt) and encodes the RNA-dependent RNA polymerase (RdRp) pivotal for the replication and transcription. The M segment, encompassing 3378 nt, encodes the glycoproteins Gn and Gc that facilitate virus attachment and invasion. Spanning 1744 nt, the S segment configures as an ambisense RNA. The genomic and antigenomic S RNA serve as the templates for the synthesis of viral mRNAs that encode the nucleoprotein (NP) and the non-structural protein (NSs), respectively. NP envelops and shields the RNA genome segments, whereas NSs represents a major viral virulence factor.

RNA modification, an complicated process of adjusting or transforming of nucleotides on RNA molecules through the addition, deletion, or alteration of chemical groups post RNA synthesis [8], emerges as a critical post-transcriptional regulatory mechanism that orchestrates diverse biological processes under both physiological and pathological conditions, including the regulation of RNA structure, stability, translation, and localization [9–11]. Recent investigations have underscored the existence of a myriad of distinctive chemical modifications in RNA molecules within viruses or host cells during viral infections, including N6-methyladenosine (m$^6$A), 5-methylcytosine (m$^5$C), N7-methylguanosine (m$^7$G), N6, 2'-O-dimethyladenosine (m$^6$Am), N1-methyladenosine (m$^1$A), N4-acetylcytidine (ac$^4$C), and 2′-O-methylated nucleosides (Nm) [12]. Despite the prolonged exploration of RNA modifications in viral RNA, only a select few, such as m$^7$G, Nm, m$^6$A, and m$^5$C, have been unveiled for their roles during the virus-host interaction [11,13,14]. Notably, m$^6$A modification, prevalent in eukaryotes, exercises regulatory dominion over various facets of eukaryotic mRNA functionality, such as alternative splicing, nuclear export, stability, and translation efficiency [15]. Moreover, evidence posits that viruses adeptly exploit m$^6$A modification to finely tune their replication cycle

[16]. Hence, the strategic targeting of m⁶A regulators may hold significant therapeutic promise in combatting viral infections.

m⁶A, a reference to the methylation modification of the nitrogen atom at the 6th position of adenine (A) in RNA, is catalyzed by m⁶A methyltransferases. Foremost among these are methyltransferase-like protein 3 (METTL3) and methyltransferase-like protein 14 (METTL14), coalescing into a vital heterodimer [17]. METTL3, *via* its methyltransferase domain, engages with S-adenosylmethionine to catalyze the conversion of A to m⁶A, while METTL14 assumes the responsibility of recognizing RNA substrates and stabilizing the conformation of METTL3—both of which are indispensable for m⁶A modification. The removal of m⁶A modification finds catalytic prowess in m⁶A demethylases, including the Fat mass and obesity-associated protein (FTO), tasked with demethylating m⁶A and m⁶Am [18], and AlkB homolog 5 (ALKBH5), an entity singularly dedicated to the demethylation of m⁶A [19]. Biological functions of RNA m⁶A modification pivot on specific m⁶A-binding proteins, such as YT521-B homology domain family proteins (YTHDF subtypes and YTHDC subtypes), insulin-like growth factor 2 mRNA-binding proteins (IGF2BPs), and heterogeneous nuclear ribonucleoproteins (HNRNPs) [20]. These discerning m⁶A-binding proteins regulate several fundamental biological processes, including RNA stability, translation, and degradation. For instance, cytoplasmic YTHDF1 amplifies mRNA translation efficiency [21], and YTHDF2 promotes mRNA decay through the CCR4-NOT complex [22], while YTHDF3 delicately modulates mRNA translation or decay through interactions with YTHDF1 or YTHDF2 [23], respectively.

Investigations have substantiated the vital role of RNA m⁶A modification in viral infections. However, during the interaction between diverse viruses and host cells, inconsistencies emerge regarding how m⁶A modification influences the denouement of viral infections. On one hand, m⁶A modification within viral RNA can directly impact the replication cycles of various viruses. As one of the evolutionarily conserved RNA modifications, prevailing studies demonstrated that m⁶A modification of viral genomic RNA promotes the infection of viruses like HIV [24], IAV [25], EV71 [26], RSV [27], HMPV [28], and SV40 [29], among others. Paradoxically, for SARS-CoV-2 [30], KSHV [31], and viruses of the Flaviviridae family (DENV, ZIKV, YFV, WNV, and HCV) [32], m⁶A modification in genomic RNA assumes a repressive role in viral infection. Mechanistically, m⁶A modification of viral RNA impacts the infection landscape by modulating processes such as RNA structure, stability, translation, cellular localization, and alternative splicing [25,27,33–35]. On the other hand, during virus-host interactions, m⁶A not only directly participates in regulating the metabolism of viral RNA but also potentially modulates viral RNA sensing, the innate immune pathway, endoplasmic reticulum stress, and cellular metabolism of host cells by influencing m⁶A modification or expression of their transcripts [36], and thus ultimately regulates viral infection. For instance, during HCMV infection, m⁶A modification targets IFN-β mRNA, accelerating its decay and, in turn, inhibiting the type I interferon pathway, thereby promoting viral infection [33,37]. Alterations in m⁶A modification of host RIOK3 and CIRBP mRNAs, induced by innate immune and endoplasmic reticulum stress signaling pathways, regulate Flaviviridae infection [38]. EBV infection downregulates m⁶A modification of KLF4 mRNA, which increases its expression and ultimately promotes the viral infection [39]. Additionally, m⁶A modification-mediated downregulation of the metabolic enzyme OGDH prompts cellular metabolic reprogramming, concomitantly hindering the infection of various model viruses such as VSV and HSV-1 [40].

In a recent longitudinal investigation of the transcriptomic and epigenetic attributes associated with SFTS patients, a potential involvement of m⁶A modification in the SFTSV genomic RNA has surfaced [41]. However, research addressing the chemical modifications of SFTSV RNA during the viral infection is still limited, and the functional implications of m⁶A

modification during SFTSV infection remain obscured. Therefore, this study aimed to elucidate the impact of RNA m$^6$A modification on SFTSV replication. Our data indicated that RNA m$^6$A modification might play a pivotal role during SFTSV infection, thereby offering a promising experimental and theoretical basis for the development of tailored anti-SFTSV pharmaceuticals, preventive vaccines, and management of tick-borne diseases.

## Result

### The SFTSV RNAs are m$^6$A methylated

We initially probed post-transcriptional modifications in SFTSV RNAs. To do this, we first extracted RNA from purified virions in the supernatant of SFTSV-infected Huh7 cells (Fig 1A), and then stringently verified its purity *via* strand-specific RT-qPCR to ensure the absence of host RNA contamination (S1A–S1C Fig). LC-MS/MS analysis [42] of viral particle RNA disclosed diverse chemical modifications, including m$^7$G, m$^5$U, m$^6$A, m$^1$A, m$^5$C, Gm, Am, and Cm (Fig 1B). Notably, around 0.32% of A bases in SFTSV RNA underwent m$^6$A methylation, aligning with levels observed in host mRNA (0.1–0.6%) [43,44].

We performed MeRIP-seq analysis in parallel using virion RNA and intracellular RNA to locate m$^6$A sites on SFTSV RNAs (Fig 1C). After mapping sequencing reads to the genome of SFTSV, we identified several m$^6$A peaks across the viral genome (Fig 1D–1F). For the genomic RNA, MeRIP-seq of virion RNA revealed m$^6$A peaks distributed across all three segments. In the S segment, two m$^6$A peaks were identified in regions complementary to the NP and NSs genes (Fig 1D). In the M segment, three m$^6$A peaks were observed in the regions complementary to the G gene (Fig 1E). In the L segment, five m$^6$A peaks were detected in the regions complementary to the RdRp gene (Fig 1F). However, further MeRIP-seq analysis of intracellular RNA from SFTSV-infected Huh7 and HeLa cells revealed distinct m$^6$A peaks in viral minus-sense RNA (Fig 1D–1F), and the m$^6$A peaks identified in intracellular minus-sense RNA were also not entirely consistent with those detected in virion genomic RNA, suggesting differential m$^6$A modification patterns between cell types and between intracellular and virion-associated RNA.

Although previous studies have reported the presence of antigenomic RNA, especially the S segment, in the virions of single-segmented negative-sense RNA (-ssRNA) viruses, such as bunyavirus Rift Valley fever virus [46,47], the presence of antigenomic RNA in SFTSV virions has not been confirmed yet. Our MeRIP-seq analysis detected m$^6$A peaks across all three segments of SFTSV virion RNA, with one m$^6$A peak located in the antigenomic RNA of each segment (S1D Fig), suggesting the potential presence of m$^6$A modifications in antigenomic RNA from virions, a phenomenon that has been observed in other -ssRNA viruses, including RSV, HMPV, and VSV [27,28,48]. Likewise, MeRIP-seq analysis of total RNA from SFTSV-infected Huh7 (S1E Fig) or HeLa cells (S1F Fig) indicated potential m$^6$A modifications in SFTSV antigenome/transcripts as well. Subsequent validation using MeRIP combined with strand-specific RT-qPCR confirmed m$^6$A modifications on genomic RNA and antigenomic RNA of S, M, and L segments. Compared to the IgG control, the m$^6$A antibody enriched regions containing m$^6$A peaks across the S, M, and L segments (Fig 1G). In conclusion, these results demonstrated the presence of m$^6$A modifications in SFTSV RNAs and suggested distinct m$^6$A modification patterns between intracellular and virion RNA, highlighting a potential role for m$^6$A in regulating different stages of SFTSV infection.

### m$^6$A promotes SFTSV infection

Until now, no intrinsic m$^6$A methyltransferase activity has been identified in the proteins encoded by SFTSV. We thus hypothesized that internal m$^6$A deposition on SFTSV RNA is

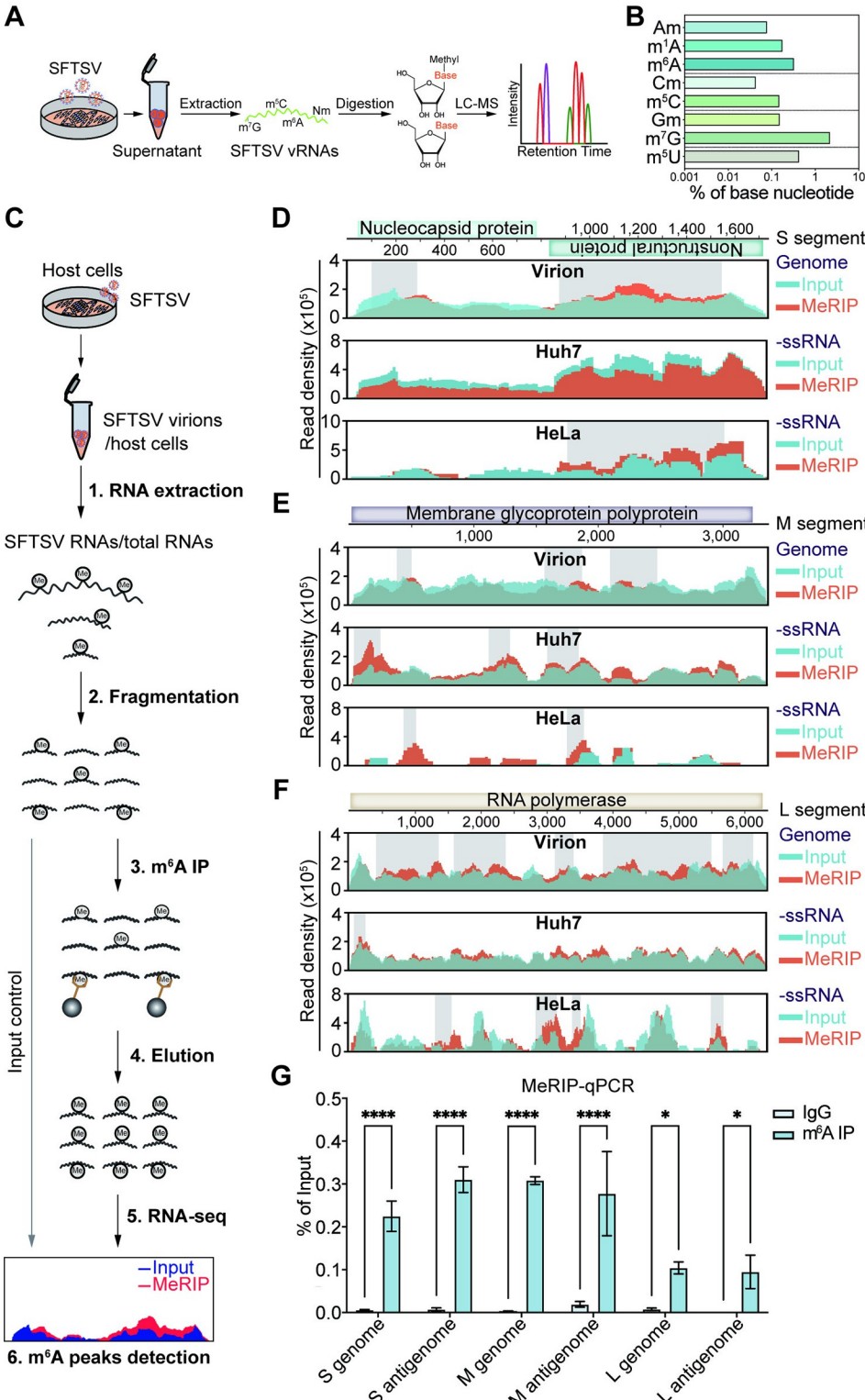

**Fig 1. The SFTSV RNAs are m⁶A methylated.** (A) Schematic diagram of LC-MS/MS analysis of SFTSV virion RNAs purified from infected Huh7 cells. (B) Levels of several different RNA modifications in SFTSV virion RNAs, expressed as percentages of the parental nucleotide, were quantified using LC-MS/MS analysis. The data were derived from a single sample. (C) Schematic diagram of MeRIP-seq. Viral RNAs/total RNAs were extracted from purified SFTSV virions/host cells, and immunoprecipitated (IP) by m⁶A-specific antibody before sequencing. (D-F) The distribution of

m$^6$A peaks in the S (D), M (E), and L (F) segments of SFTSV genome/-ssRNA. MeRIP-seq mapping of SFTSV RNAs derived from either purified virions or SFTSV-infected Huh7 or HeLa cells. Genome/-ssRNA: The baseline signal of Input was displayed as cyan, and the m$^6$A IP signal was displayed as orange. The gray rectangle indicated the m$^6$A peaks recognized by MACS2 [45]. (G) The presence of m$^6$A modification in SFTSV RNAs was verified by MeRIP-qPCR. Data are representative of three independent experiments and presented as mean ± SD. Statistical significance was determined by two-way ANOVA followed by Sidak's multiple comparisons test. *, $P < 0.05$; ****, $P < 0.0001$.

facilitated by the host METTL3-METTL14 heterodimer, the core of the m$^6$A methyltransferase complex [17]. To investigate the impact of m$^6$A methyltransferase in SFTSV infection, shRNAs targeting METTL3 or METTL14 were constructed (S2A and S2B Fig). After silencing METTL3 or/and METTL14, we observed a substantial reduction in viral RNA load, the relative amounts of viral RNA released in the supernatant, and NP protein expression in SFTSV-infected Huh7, HeLa, and HEK293T cells (Fig 2A). Conversely, METTL3 or/and METTL14 overexpression produced opposite results (Fig 2B). Overall, these results demonstrated that m$^6$A methyltransferase enhances SFTSV replication, and thereby augments viral protein expression and progeny virus load. As such, m$^6$A modification appears to facilitate SFTSV infection.

m$^6$A modification is dynamically reversible and can be demethylated in a highly regulated manner by m$^6$A demethylases. We then wondered whether m$^6$A demethylases regulate SFTSV infection as well. Using specific shRNAs targeting m$^6$A demethylases (S2C and S2D Fig), we found that ALKBH5 knockdown resulted in significant increases in the cellular viral RNA load, the relative amounts of viral RNA released in supernatant, and the expression of NP protein in Huh7, HeLa, and HEK293T cells infected with SFTSV (Fig 2C). However, silencing FTO alone showed little effect (Fig 2C). Conversely, overexpression of ALKBH5 alone or in combination with FTO significantly reduced intracellular and extracellular SFTSV RNA load, as well as the NP protein expression (Fig 2D). Thus, ALKBH5 inhibits SFTSV life cycle in infected cells, reinforcing the role of m$^6$A in promoting SFTSV infection.

To ascertain whether m$^6$A directly promotes SFTSV infection, we tested whether METTL3 or ALKBH5 regulates SFTSV infection *via* their m$^6$A catalytic activities. As shown in Fig 2E, the METTL3 APPA mutation [49] or ALKBH5 H204A mutation [19] largely abrogated the ability of their corresponding WT protein to promote or diminish cellular SFTSV RNA loads and NP protein levels. These results strongly indicated that METTL3 or ALKBH5 regulates the lifecycle of SFTSV through their m$^6$A catalytic activities. Furthermore, METTL3 knockout in Huh7 cells by CRISPR/Cas9 resulted in a decreased virus titer (Fig 2F), lower levels of SFTSV RNA levels (Fig 2G), and reduced NP protein levels (Fig 2H), compared to wild-type (WT) cells. Additionally, METTL3 knockout led to a significant reduction in m$^6$A levels in both cellular RNA and SFTSV virion RNA (Fig 2I). Therefore, these findings provide further evidence that m$^6$A modification enhances SFTSV infection.

Considering that SFTSV replicates in the cytoplasm where m$^6$A-binding proteins YTHDF1/2/3 locate, and that YTHDF1/2/3 broadly promote virus infection in an m$^6$A-dependent manner [25,27], we reasoned that YTHDF1/2/3 proteins may have a similar impact on the SFTSV lifecycle. Indeed, silencing endogenous YTHDF1/2/3 proteins decreased, while their overexpression increased, the intracellular and extracellular SFTSV RNA levels and NP protein expressions (S2E–S2G and S3 Figs). Thus, these results indicated that YTHDF1/2/3 proteins phenocopy the m$^6$A methyltransferases in terms of facilitating SFTSV replication and viral protein expression.

Employing the CRISPR-Cas13 RNA editing system [50], we tested the impact of m$^6$A modification in SFTSV genomic RNA on SFTSV replication. Using dCas13b-ALKBH5/gRNA targeting demethylation system, we specifically reduced m$^6$A modification levels on SFTSV genomic RNA in SFTSV-infected Huh7 cells (S4A Fig). MeRIP-RT-qPCR results showed that

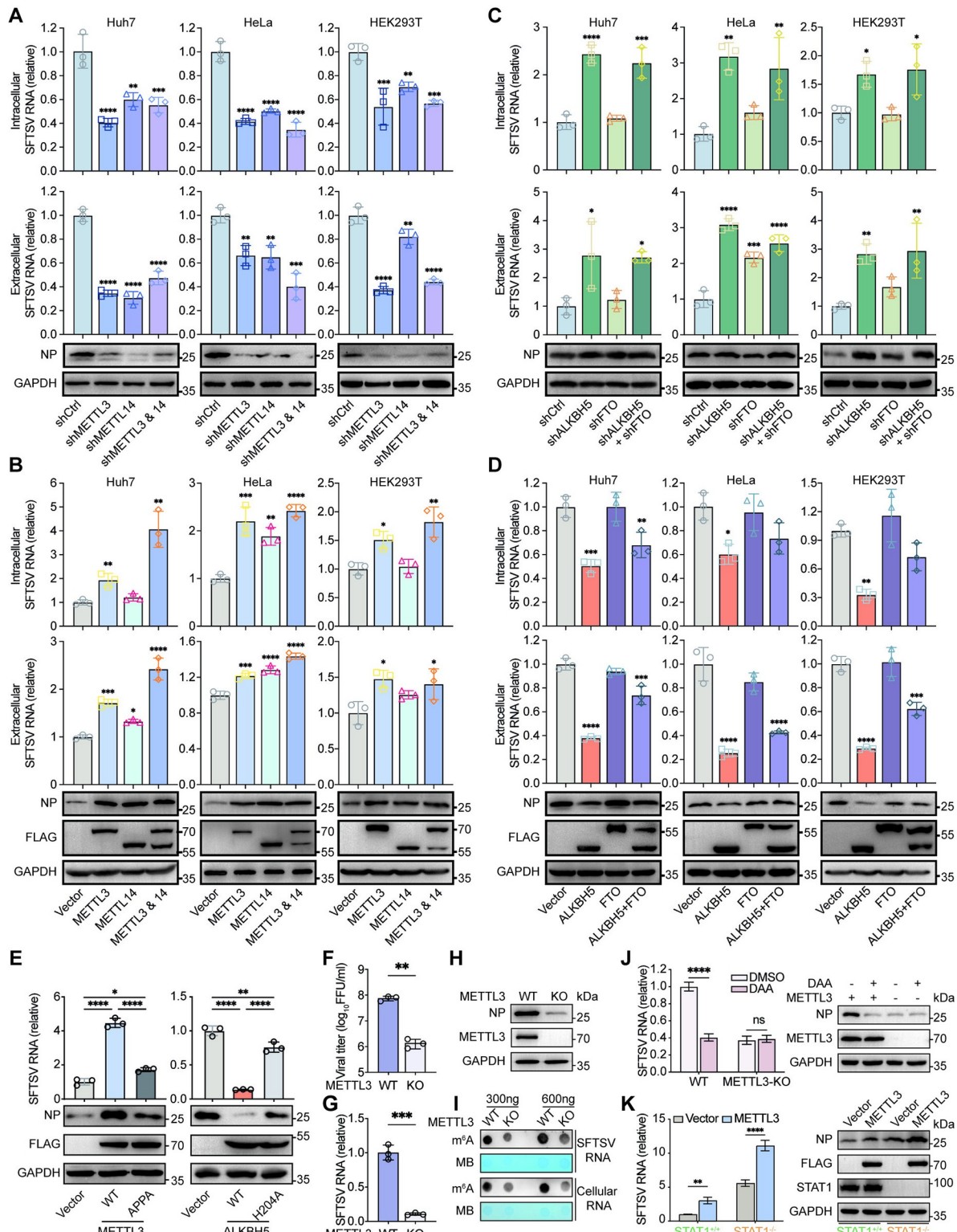

**Fig 2. m⁶A promotes SFTSV infection.** (A-D) Effects of m⁶A methyltransferases (A-B) and demethylases (C-D) on SFTSV infection. Huh7, HeLa and HEK293T cells were infected with lentiviral particles containing control (shCtrl), shRNAs targeting METTL3/METTL14 (A) or ALKBH5/FTO (C), followed by puromycin selection. Alternatively, cells were transfected with the empty vector or plasmids encoding indicated m⁶A regulators (B and D) for 24 h. Then these m⁶A regulators-depleted or -overexpressed cells were infected with SFTSV (MOI = 1) for 48 h. The relative intracellular SFTSV RNA levels were quantified by RT-qPCR using ACTB as the reference gene,

while the released SFTSV RNA isolated from equal volumes of supernatants were quantified by RT-qPCR and normalized to the control group. NP protein levels were detected by western blot, with GAPDH as the loading control. (E) The catalytic activity of METTL3 or ALKBH5 is critical for regulating SFTSV infection. Huh7 cells overexpressing WT/APPA-mutated METTL3 (left) or WT/H204A-mutated ALKBH5 (right) were infected with SFTSV for 48 h, and relative intracellular SFTSV RNA levels and protein levels of METTL3/ALKBH5 and NP were measured. (F-H) Knockout of METTL3 reduces SFTSV infection. (F) SFTSV titer by Focus-forming Assay (FFA) in supernatants of wide-type (WT) or METTL3-deficient (KO) Huh7 cells infected with SFTSV for 48 h. (G) RT-qPCR of SFTSV RNA or (H) western blot of SFTSV NP protein of METTL3 KO or WT Huh7 cells infected with SFTSV for 48 h. (I) Dot blot of m6A levels (300 ng/600 ng RNA) in virion RNA and cellular RNA from WT and METTL3 KO Huh7 cells infected with SFTSV for 48 h. Methylene blue (MB) staining served as a loading control. (J) Effect of DAA on SFTSV infection in METTL3 KO cells. RT-qPCR analysis of SFTSV RNA expression or western blot analysis showing NP protein levels in SFTSV-infected (MOI = 1) WT or METTL3 KO cells treated with or without DAA (10 μM) at 48 hpi. (K) Effect of METTL3 overexpression on SFTSV infection in STAT1 KO cells. RT-qPCR analysis of SFTSV RNA expressions or western blot of NP protein levels in STAT1-sufficient or deficient cells infected with SFTSV (MOI = 1) for 48 h. Data are representative of three independent experiments and presented as mean ± SD. Statistical significance was determined by one-way ANOVA followed by Dunett's multiple comparisons test (A-D), one-way ANOVA followed by Tukey's multiple comparisons test (E), student's t test (F and G), or two-way ANOVA followed by Sidak's multiple comparisons test (J and K). *, $P < 0.05$; **, $P < 0.01$; ***, $P < 0.001$; ****, $P < 0.0001$. MTD, methyltransferase domain; DSBH, double-stranded β-helix.

the reduced m6A modification levels in S, M, or L genomic RNA were concurrent with the decreased intracellular SFTSV RNA load (S4B–S4E Fig), suggesting that m6A modification in viral genomic RNA facilitates SFTSV infection.

3-Deazaadenosine (DAA) is a potent inhibitor of internal m6A modification [51]. To test the impact of DAA on SFTSV infection, we treated Huh7 cells with various concentrations of DAA, and found that DAA, at the range of 0–50 μM, significantly reduced the cellular SFTSV RNA load and NP protein expression in a dose-dependent manner without affecting cellular viabilities (S4F and S4G Fig). However, these inhibitory effect of DAA on SFTSV infection was nearly abolished in METTL3 KO cells (Fig 2J). These results indicated that the SFTSV lifecycle is highly sensitive to the inhibition of cellular methylation.

Studies have proposed that m6A may promote viral infection by interfering with the type I interferon pathway, as it interferes with the recognition of viral RNA by RIG-I [28] or targets IFN-β mRNA for accelerated degradation [33,37]. Notably, we found that overexpression of METTL3 promoted SFTSV infection in STAT1 KO cells (Fig 2K). Thus, these findings suggest that the role of m6A in promoting SFTSV infection is not exclusively dependent on the type I interferon pathway.

## SFTSV infection alters the expression profiles of m6A regulators

A quantitative proteomic study utilizing isobaric tags for relative and absolute quantitation (iTRAQ) technology has revealed an upregulation of host m6A regulators in SFTSV-infected HEK293 cells (Fig 3A) [52]. Likewise, by reanalyzing the transcriptomic sequencing data from blood cells of SFTS patients (GSE144358) [53], we found that the transcriptional levels of most m6A methyltransferases and reader proteins were increased in acute deceased SFTSV infection group as compared to those in the healthy control group (Fig 3B). In keeping with these findings, the mRNA levels of METTL3 & METTL14 (m6A writers), ALKBH5 & FTO (m6A erasers), and YTHDF2 (m6A reader) were significantly increased in a time-dependent fashion in Huh7 cells post SFTSV infection (Fig 3C). Furthermore, we observed upregulation of METTL3 and METTL14 proteins during SFTSV infection, accompanying with increased cellular m6A level in total RNA (Fig 3D). Collectively, these results indicated that SFTSV infection augments the expression of most m6A regulators at both mRNA and protein levels.

## SFTSV nucleoprotein interacts with METTL3 and alters its cellular distribution

Given the fact that SFTSV NP binds to both SFTSV genomic RNA and anti-genomic RNA, we hypothesized that NP interacts with host m6A regulators to modulate the m6A modification of

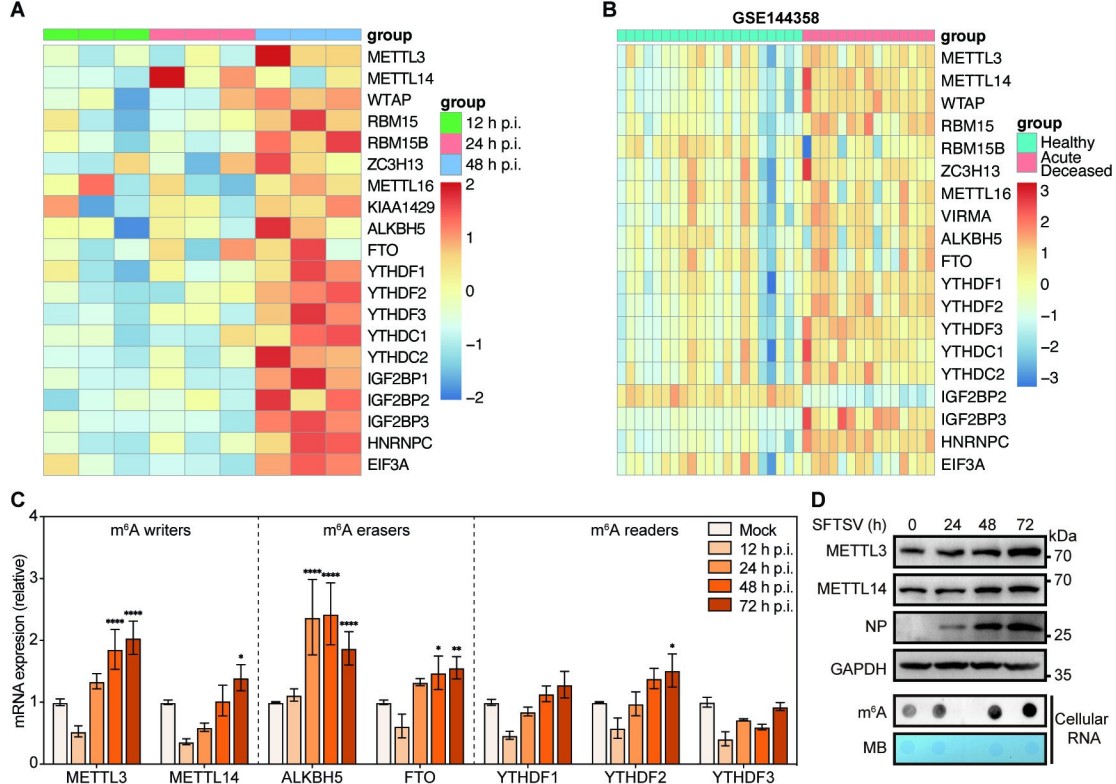

**Fig 3. The expression profiles of m⁶A regulators during SFTSV infection.** (A) Heatmap profiling of m⁶A regulators expressions from iTRAQ-based quantitative proteomic analysis of SFTSV-infected HEK293 cells [52]. (B) Heatmap profiling of m⁶A-related gene expressions from transcriptomic analysis of blood cells isolated from 23 healthy controls and 17 Acute Deceased SFTS patients in GSE144358 datasets [53]. (C-D) Time course analysis of m⁶A-related gene expressions in SFTSV-infected Huh7 cells. (C) Total RNA collected from SFTSV-infected cells (MOI = 1) at the indicated hours post infection (hpi). For each indicated gene, mRNA was analyzed by RT-qPCR and normalized to ACTB. (D) Western blot of METTL3 and METTL14 protein levels and Dot blot of m⁶A levels (300 ng total RNA) in Huh7 cells infected with SFTSV as indicated times. Methylene blue (MB) staining served as a loading control. Data are representative of three independent experiments and presented as mean ± SD. Statistical significance was determined by one-way ANOVA followed by Dunett's multiple comparisons test (C). *, $P < 0.05$; **, $P < 0.01$; ****, $P < 0.0001$.

viral RNA. Therefore, we employed immunoprecipitation-mass spectrometry analysis (IP-MS) to screen host proteins that may interact with SFTSV NP. Not surprisingly, the IP-MS results identified several potential NP-interacting m⁶A regulators, among which METTL3 stood out as the top hit in the list (Figs 4A and S5A). Our exogenous Co-IP assay confirmed that METTL3 not only interacts with SFTSV NP, as evidenced by their reciprocal detection in each other's immunoprecipitates (Fig 4B), but also interacts with other SFTSV protein such as G protein (S5B Fig). Moreover, endogenous Co-IP further validated the METTL3-NP interaction in SFTSV-infected Huh7 cells (S5C Fig), and confocal microscopy analysis also revealed that SFTSV infection resulted in the translocation of endogenous or exogenous METTL3 from the nucleus to the cytoplasm, where it colocalized with the SFTSV NP (Figs 4C and S5D). Accordingly, nuclear-cytoplasmic fractionation combined with WB analysis showed an increase of cytoplasmic METTL3 both in Huh7 cells (Fig 4D) and HEK293T cells (S5E Fig) upon SFTSV infection. Given that both METTL3 and NP are RNA-binding proteins, we performed RNase treatment before Co-IP to determine whether the interaction between METTL3 and NP is mediated by RNA in a non-specific manner. Our result showed that

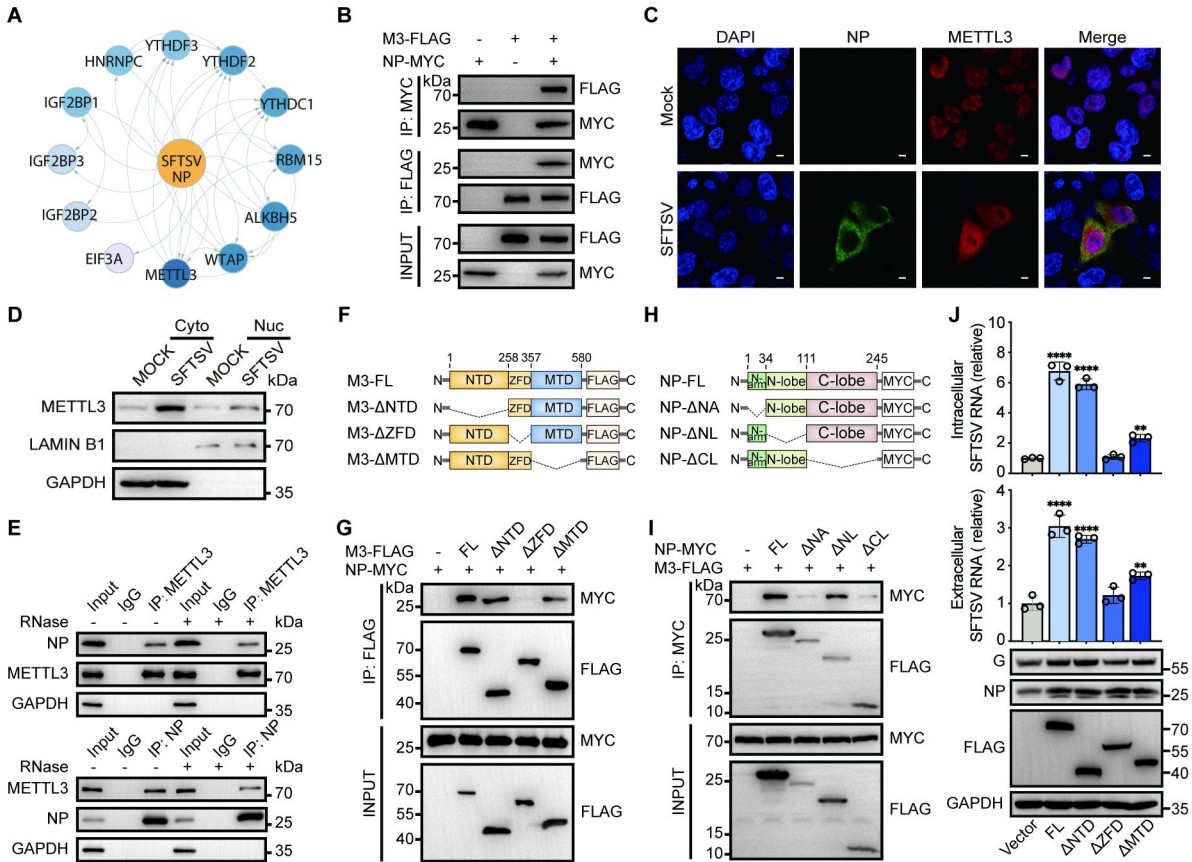

**Fig 4. The interaction between the zinc finger domain of METTL3 and the C-lobe domain of SFTSV NP is required for facilitating SFTSV infection.** (A) Protein-protein interaction (PPI) network representing the potential interaction among SFTSV NP and m⁶A regulators. HEK293T cells were transfected with NP-FLAG for 24 h, followed by SFTSV infection (MOI = 1). Immunoprecipitation was performed with anti-Flag antibody-coupled magnetic beads 36 hpi to identify potential NP-interacting m⁶A regulators by mass spectrometry. (B) METTL3 interacts with SFTSV NP. The exogenous interaction between METTL3 and SFTSV NP was detected by Co-IP assay. HEK293T cells were co-transfected with plasmids encoding FLAG-tagged METTL3 and MYC-tagged NP for 36 h and cell extracts were subjected to immunoprecipitation (IP) assay using anti-FLAG or anti-MYC antibody, respectively. Whole-cell lysates and the immunoprecipitates were probed with anti-MYC or anti-FLAG antibody, respectively. (C-D) SFTSV infection alters the distribution of METTL3. (C) Representative confocal microscopy images of mock- or SFTSV-infected HeLa cells (48 hpi, MOI = 1). The nucleus (blue), METTL3 (red) and NP protein (green) were labeled with DAPI, anti-METTL3, and anti-NP antibodies, respectively. Scale bar, 10 μm. (D) Nuclear and cytosolic fractions of mock- or SFTSV-infected Huh7 cells (48 hpi, MOI = 1) were separated and probed with anti-METTL3 antibody. GAPDH/LAMIN B1 were used as cytoplasmic/nuclear markers, respectively. (E) METTL3 interacts with SFTSV NP in an RNA-independent manner. SFTSV-infected Huh7 cell extracts treated with or without RNase A were subjected to IP analysis using the control IgG, anti-METTL3 or anti-NP antibodies. The immunoprecipitates were analyzed with antibodies indicated on the left side. (F-G) The ZFD of METTL3 interacts with SFTSV NP. Interactions between METTL3 truncations (illustrated in F) and NP in HEK293T cells were detected by Co-IP assay (G). (H-I) The C-lobe domain of NP interacts with METTL3. Interactions between NP truncations (shown in H) and METTL3 in HEK293T cells were detected by Co-IP assay (I). (J) The ZFD of METTL3 is required for promoting SFTSV infection. Huh7 cells overexpressing the full length or different truncations of METTL3 were infected with SFTSV (MOI = 1) for 48 h. Relative intracellular SFTSV RNA levels were quantified by RT-qPCR at 48 hpi with ACTB as the reference gene, while the released SFTSV RNA isolated from equal volumes of supernatants were quantified by RT-qPCR and normalized to the control group. The G and NP protein levels were analyzed by western blotting. Data are representative of three independent experiments and presented as mean ± SD. Statistical significance was determined by one-way ANOVA followed by Dunett's multiple comparisons test. **, $P < 0.01$; ****, $P < 0.0001$.

METTL3 still interacts with SFTSV NP after RNase treatment (Fig 4E), indicating the interaction is RNA-independent. In summary, these results indicated that SFTSV may recruit the host m⁶A methyltransferase METTL3 through NP to promote the m⁶A modification of its own viral RNA.

## The zinc finger domain of METTL3 interacts with the C-Lobe domain of SFTSV NP and is required for facilitating SFTSV infection

METTL3 is characterized by several essential domains including the N-terminal WTAP binding domain (NTD), the zinc finger domain (ZFD) (aa 259–336) [54,55], and the C-terminal methyltransferase domain (MTD) (aa 357–580) [49,54]. To examine which domain of METTL3 interacts with NP, we constructed different truncated forms of METTL3 and tested their potential interactions with NP using Co-IP assays (Fig 4F). Results showed that deletion of the ZFD (ΔZFD) abolished co-immunoprecipitation of MELLT3 with SFTSV NP, while deletion of the NTD (ΔNTD) or MTD (ΔMTD) was ineffective (Fig 4G), indicating that the ZFD of METTL3 mediates its interaction with SFTSV NP.

Likewise, SFTSV NP comprises three crucial domains: the N-terminal arm (N-ARM) (M1-D34), the N-lobe domain (P35-L111), and the C-lobe domain (P112-L245) [56]. We next took a similar approach to determine which fragment of SFTSV NP (Fig 4H) mediates its interaction with METTL3, and found that fragments lacking either the N-ARM (ΔNA) or the C-lobe domain (ΔCL) largely abrogated the co-immunoprecipitation of SFTSV NP with METTL3 (Fig 4I). Given that the N-arm is essential for the oligomerization of SFTSV NP and deletion of N-ARM demolishes its stable hexameric structure [56], we speculated that the C-lobe domain of SFTSV NP interacts with the ZFD of METTL3.

Coinciding with their NP-binding abilities (Fig 4F and 4G), the full-length METTL3, ΔNTD, and ΔMTD, rather than ΔZFD, were able to enhance SFTSV infection (Fig 4J), indicating that ZFD-mediated METTL3-NP interaction is indispensable for augmented SFTSV lifecycle. Intriguingly, the ΔMTD truncated form of METTL3 lacking the critical catalytic site (DPPW motif) enhanced SFTSV replication as well (Fig 4J), which may be related to the non-m$^6$A-dependent translational enhancement ability of METTL3 as previously reported [57]. Nevertheless, the ability of ΔMTD truncation to promote SFTSV infection was significantly weaker than that of the full-length METTL3 or the ΔNTD truncated form, aligning with our previous observations that the METTL3 APPA mutation impaired the ability of METTL3 to promote SFTSV infection (Fig 2E). Thus, METTL3 appears to promote SFTSV infection both m$^6$A dependently and independently.

## METTL3 orchestrates m$^6$A modification of NP mRNA to enhance its translation efficiency

We have showed that SFTSV infection induces the translocation of METTL3 from the nucleus to the cytoplasm (Fig 4C and 4D), where it may interact with the translation initiation complex and thus enhances translation [58]. We therefore hypothesized that METTL3 regulates the translation of SFTSV transcripts. To test this hypothesis, we co-transfected NP- and METTL3-encoding plasmids into HEK293T cells to avoid the effects of viral vRNA or cRNA on the expression of viral proteins. RIP-qPCR showed that METTL3 significantly enriched NP mRNA compared to the IgG control (Fig 5A), indicating the binding of NP transcript by METTL3. Despite the comparable NP transcript expressions (Fig 5B), METTL3 overexpression increased NP protein levels in a dose-dependent manner (Fig 5C), suggesting that METTL3 may facilitate NP transcript translation. To further confirm this, we constructed a pmirGLO-NP-CDS dual-luciferase reporter gene. Dual-luciferase assays showed that METTL3-overexpression indeed significantly increased the NP translation efficiency in HEK293T cells (Fig 5D).

We next utilized the SRAMP database [60] to predict m$^6$A sites in the NP CDS that may affect its expression. A68 had low confidence whereas A273 had extremely high confidence (Fig 5E). We disrupted the consensus m$^6$A motif of these sites by synonymous mutations and

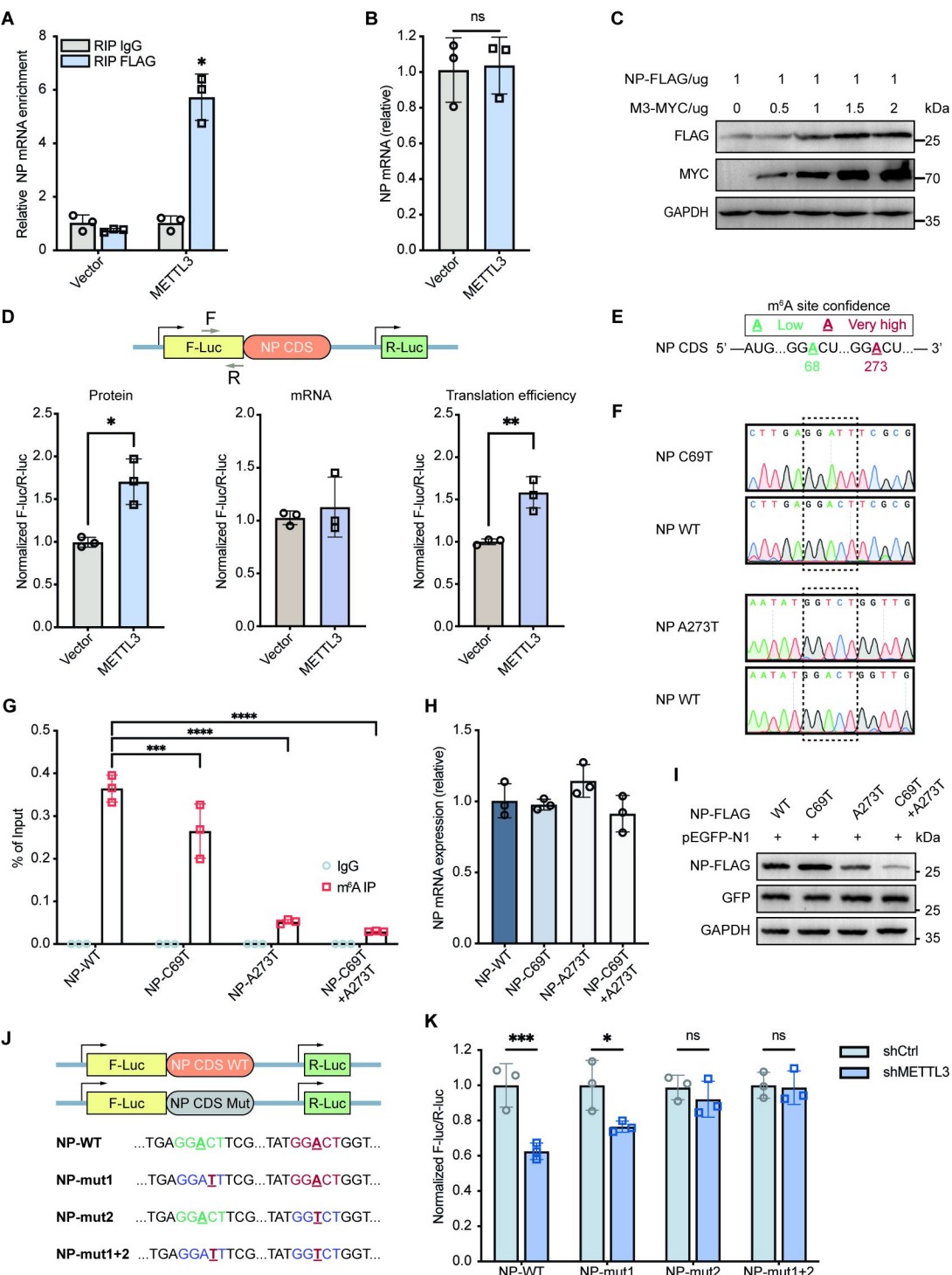

**Fig 5. METTL3 orchestrates m⁶A modification of NP mRNA to enhance its translation efficiency.** (A) The interaction between METTL3 and NP mRNA was analyzed by RIP-qPCR assay in HEK293T cells with METTL3 and NP overexpressions. (B) NP mRNA expression levels were detected by RT-qPCR in METTL3- and NP- overexpressed HEK293T cells. (C) METTL3 promotes NP protein expressions. HEK293T cells were transfected with 1 μg NP-FLAG together with various amounts of METTL3-MYC (0–2 μg) in six-well plates. Expressions of METTL3 and NP were determined by western blotting. (D) METTL3 augments the translation efficiency of NP mRNA. HEK293T cells were co-transfected with METTL3-FLAG and pmirGLO-NP-CDS reporter for 24 h. The translation outcome was determined as a relative signal of F-luc divided by R-luc, and the mRNA abundance was determined by RT-qPCR of F-luc and R-luc. The translation efficiency of NP is defined as the

quotient of reporter protein production (F-luc/R-luc) divided by mRNA abundance [21, 59].(E-F) Schematic diagram of predicted m⁶A sites and synonymous mutations introduced in the NP CDS. m⁶A sites in the CDS sequence of NP were predicted by SRAMP program [60], with green/red indicating low/very high confidence, respectively. (G) The interaction between anti-m⁶A antibody and NP mRNA was analyzed by RIP-qPCR assay in HEK293T cells transfected with NP plasmids containing either WT or mutated m⁶A sites. (H) NP mRNA expression levels were detected by RT-qPCR in HEK293T cells with ectopic expressions of NP-WT or NP mutants. (I) NP protein levels were determined by western blot in HEK293T cells transfected with pEGFP-N1 together with the pCAGGS vectors encoding NP-WT or NP mutants. Endogenous GAPDH and GFP levels were used as controls for loading and transfection, respectively. (J) Schematic representation of predicted m⁶A sites and synonymous mutations introduced in the NP CDS of pmirGLO vector. (K) Dual luciferase reporter assays showing the effect of shMETTL3 on NP reporters with either WT or mutated m⁶A sites. Data are representative of three independent experiments and presented as mean ± SD. Statistical significance was determined by two-way ANOVA followed by Sidak's multiple comparisons test (A, G and K), student's t test (B and D) or one-way ANOVA followed by Dunett's multiple comparisons test (H). ns, not significant; *, $P < 0.05$; **, $P < 0.01$; ***, $P < 0.001$.

constructed pCAGGS-NP-C69T, pCAGGS-NP-A273T, and pCAGGS-NP-C69T+A273T (Fig 5F). The A273T mutation significantly reduced NP mRNA enrichment by the anti-m⁶A antibody in transfected HEK293T cells, and a further slight reduction was observed in cells transfected with NP-C69T+A273T (Fig 5G), indicating a crucial role of A273 in NP expression. We then measured NP mRNA levels in these plasmid-transfected cells and found that NP mRNA levels did not differ significantly between WT and m⁶A mutant forms (Fig 5H). However, NP protein levels were significantly reduced in cells expressing the A273T mutant or C69T +A273T mutant, whereas C69T had no visible impact (Fig 5I). We also created reporter constructs with WT or m⁶A-mutated NP CDS (NP-mut1, NP-mut2, or NP-mut1+2) (Fig 5J). As expected, METTL3 knockdown significantly reduced luciferase activities of the pmir-GLO-NP-WT reporter, while this reduction was completely abolished when the A273 was mutated (NP-mut2 in Fig 5K), suggesting that MELLT3-mediated m⁶A modification at the A273 site promotes NP mRNA translation efficiency and leads to augmented NP protein expression.

## METTL3-mediated m⁶A modification of SFTSV RNA affects its stability

Previous studies have demonstrated that internal m⁶A modification of eukaryotic transcripts regulates mRNA stability [22]. Our IP-MS data identified several m⁶A-binding proteins, including IGF2BPs with roles in RNA stability maintenance [61], as candidate interactors of the SFTSV NP (Fig 4A). Subsequent confocal and Co-IP experiments confirmed that SFTSV NP interacted with the m⁶A-binding proteins YTHDF1/3 (S5F–S5K Fig) and IGF2BP1/2/3 (S5L–S5Q Fig). Moreover, METTL3 silencing in Huh7 cells reduced m⁶A modification and relative expression levels of SFTSV L/M/S genomic RNA and anti-genomic RNA (Fig 6A and 6B).

We therefore reasoned that METTL3 promoted the m⁶A modification and stability of SFTSV RNA. To this end, we transfected METTL3-silenced HEK293T cells with pmirGLO-S-genome or pmirGLO-S-antigenome reporters containing the S genome or S antigenome sequence, and then measured the half-life of Fluc-S-genome or Fluc-S-antigenome mRNA. In the presence of transcription inhibition by Actinomycin D, knockdown of METTL3 shortened the half-life of Fluc-S-genome (from 26.54 h to 10.45 h) and Fluc-S-antigenome (from 6.11 h to 4.45 h) mRNA (Fig 6C and 6D). Thus, these results suggested that METTL3 enhances the stability of SFTSV S RNA, possibly *via* IGF2BP1/2/3 proteins recruited by SFTSV NP.

## m⁶A mutations in S genome diminish virus particle production

We used the SRAMP program [60] to look for potential m⁶A sites on the S genomic RNA that may affect its stability. The results revealed 15 potential m⁶A sites on the S genomic RNA (Fig 6E). To identify the functional m⁶A sites modulating S vRNA stability, we screened gRNAs

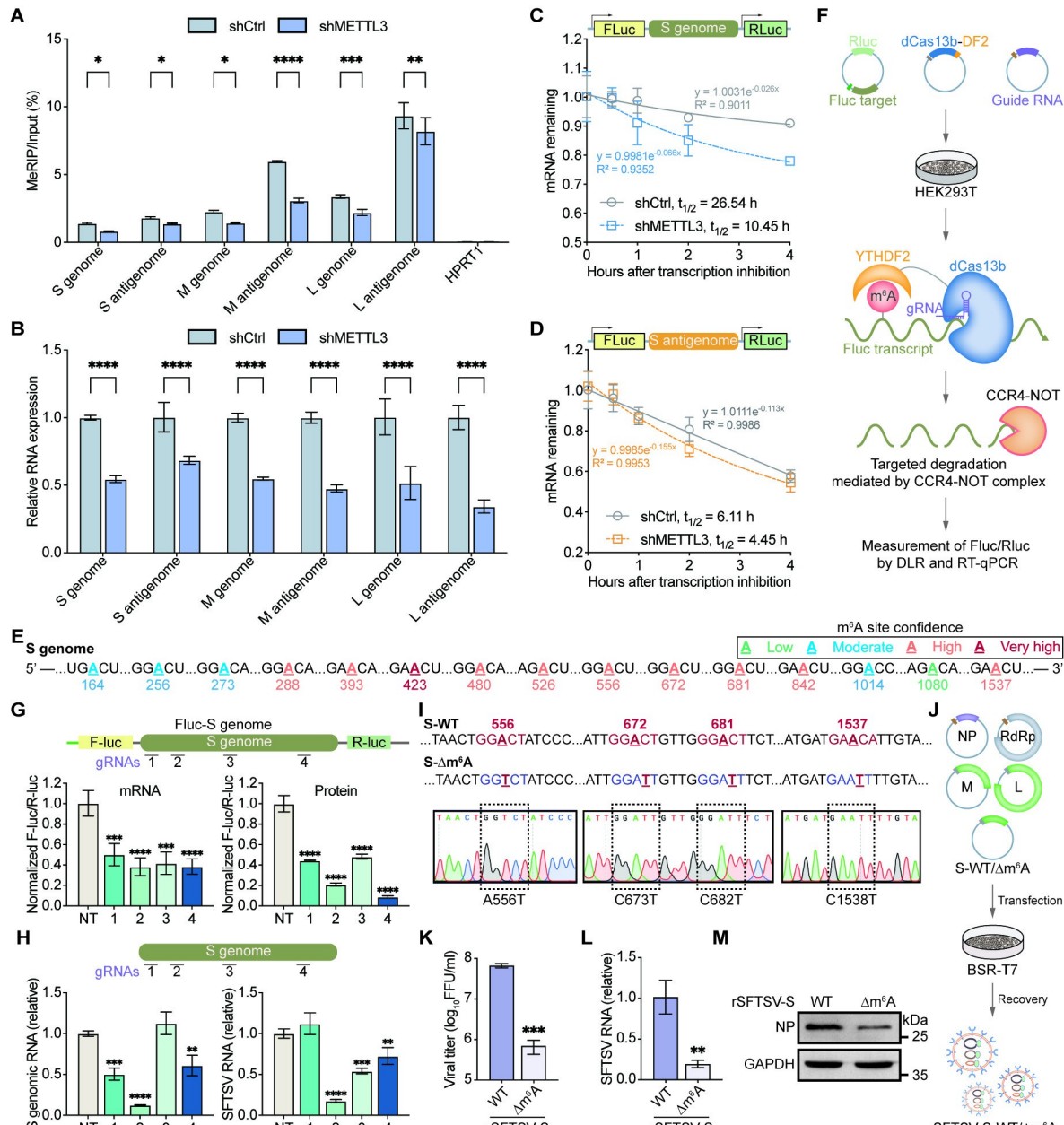

**Fig 6. METTL3-mediated m⁶A modification affects the stability of SFTSV RNA.** (A-B) The effect of METTL3 knockdown on m⁶A modifications of SFTSV RNA. METTL3-downregulated Huh7 cells were infected with SFTSV at an MOI of 1 for 48 h. The m⁶A modifications on SFTSV S/M/L genome and antigenome RNAs (A) and expressions of different SFTSV RNAs (B) were assessed with anti-m⁶A IP followed by strand-specific RT-qPCR. HPRT1 was used as a negative control. (C-D) Effects of METTL3-silencing on the stability of Fluc-S-genome/Fluc-S-antigenome mRNA in Huh7 cells. The mRNA half-life ($t_{1/2}$) of Fluc-S-genome (C) or Fluc-S-antigenome (D) transcripts in Act D-treated Huh7 cells with (shMETTL3) or without (shCtrl) METTL3 depletion. (E) A schematic illustration of predicted m⁶A sites in the S genome or antigenome by SRAMP program [60], with green, blue, orange, and red indicating the low, moderate, high, and very high confidence, respectively. (F) Schematic of gRNA screen with dCas13-YTHDF2 targeting assay. dCas13-YTHDF2, guide RNA, and a dual luciferase plasmid were transfected into HEK293T cells prior to the measurement of luciferase RNA abundance and luminescence. Firefly luciferase (Fluc, brown) reporter transcripts targeted by dCas13-YTHDF2 fusion proteins may undergo the CCR4-NOT complex-mediated degradation, with Renilla luciferase (Rluc, green) acting as a normalization control. (G) dCas13b targeting the SFTSV S genome/antigenome RNA attached to a Fluc reporter. Indicated dCas13b guide RNAs were tiled across each reporter. (H) Endogenous SFTSV genomic RNA targeting with dCas13b-YTHDF2. SFTSV-infected Huh7 cells were transfected with plasmids encoding dCas13b-YTHDF2 fusion protein and dCas13b guide RNAs that target different sites of SFTSV genome. Knockdown efficiencies of SFTSV genomic RNA as well as viral RNA load, mediated by dCas13b-YTHDF2 in combination with different gRNAs, were quantified with strand-specific RT-qPCR and RT-qPCR, respectively, to identify the regions with most robust effect. (I) Schematic of the consensus m⁶A

motifs (red) and synonymous mutations generated in the SFTSV S genome of pcDNA3.1(+) vector. (J) Procedure overview for generating rSFTSV viruses. Recovery of rSFTSV viruses was achieved by co-transfection of pCAGGS-NP, pCAGGS-RdRp, pcDNA3.1(+)-L, pcDNA3.1(+)-M, and pcDNA3.1(+)-S-WT/$\Delta$m$^6$A into BSR-T7 cells. (K) Viral titer by focus-forming assay (FFA) in supernatants of Huh7 cells infected with rSFTSV-S-WT/$\Delta$m$^6$A (MOI = 10) at 96 hpi. (L) RT-qPCR quantification of intracellular SFTSV RNA using specific primers annealing to the SFTSV G gene. (M) Immunoblot analysis of SFTSV NP protein levels in cell lysates harvested at 96 hpi with GAPDH as a loading control. Data are representative of three independent experiments and presented as mean ± SD. Statistical significance was determined by two-way ANOVA followed by Sidak's multiple comparisons test (A and B), one-way ANOVA followed by Dunett's multiple comparisons test (G and H), or student's t test (K and L). NT, non-targeting guide RNA; *, $P < 0.05$; **, $P < 0.01$, ***, $P < 0.001$; ****, $P < 0.0001$.

against S vRNA in the Fluc-S-vRNA 3'-UTR using the dCas13b-YTHDF2 RNA targeting system [62] (Fig 6F). Dual luciferase assay and RT-qPCR analysis identified that gRNA2 and gRNA4 were potential targets with most pronounced effect (Fig 6G). Moreover, co-transfection of gRNA2 or gRNA4 with dCas13b-YTHDF2 expressing vector significantly reduced both the endogenous cellular S genomic RNA level and viral RNA load in SFTSV-infected Huh7 cells (Fig 6H), indicating a crucial role of the m$^6$A sites in proximity of the regions targeted by gRNA2 or gRNA4 (A556, A672, A681, and A1537).

Brennan et al. [63] and Xu et al. [64] previously had constructed a reverse genetic system based on the SFTSV strain. We thus mutated the consensus m$^6$A sequence of all four m$^6$A sites (A556, A672, A681, and A1537) without changing the encoded amino acids to elucidate their significance in the SFTSV life cycle (Fig 6I). Subsequently, we co-transfected BSR-T7 cells with pCAGGS-NP, pCAGGS-RdRp, pcDNA3.1(+)-T7-L, pcDNA3.1(+)-T7-M, and pcDNA3.1(+)-T7-S-WT/$\Delta$m$^6$A to obtain the infectious rSFTSV viruses with wild-type or m$^6$A mutant S genome (Fig 6J). To compare the infectivity differences between wild-type and m$^6$A mutated rSFTSV, we infected Huh7 cells with rSFTSV-S-WT or rSFTSV-S-$\Delta$m$^6$A at an MOI of 10. Focus-forming assays revealed a significant decrease in the titer of rSFTSV-S-$\Delta$m$^6$A compared to that of rSFTSV-S-WT (Fig 6K). Additionally, rSFTSV-S-$\Delta$m$^6$A also synthesized less SFTSV RNA (Fig 6L) and NP protein (Fig 6M) in Huh7 cells. Collectively, these findings demonstrated that m$^6$A modification in rSFTSV enhances the viral replication and propagation in a cell infection model.

## m$^6$A-deficient rVSV-SFTSV-G exhibits replication defects *in vitro* and *in vivo*

Similarly, we used the SRAMP database [60] to identify potential m$^6$A sites essential for the stability of SFTSV M genomic RNA as well. The results identified 28 potential m$^6$A sites, with 9, 8, 8, and 2 sites having low, moderate, high, and extremely high confidence, respectively (Fig 7A). Using the dCas13b-YTHDF2/gRNA system, we targeted different m$^6$A sites to compare their roles in maintaining the stability of endogenous M genomic RNA in SFTSV-infected cells. Strand-specific RT-qPCR results showed that gRNA2, gRNA5, and gRNA1 were most efficient in degrading the M genomic RNA, suggesting that A494, A1389, and A1943 were important for its stability (Fig 7B). Moreover, we mutated the m$^6$A consensus sequence in the indicated m$^6$A sites by introducing synonymous mutations (Fig 7C). Like rSFTSV-S, we also obtained the infectious rSFTSV-G viruses with wild-type or m$^6$A mutant G gene. Similarly, Huh7 cells infected with rSFTSV-G-WT or rSFTSV-G-$\Delta$m$^6$A at an MOI of 10 showed a significant decrease in the titer of rSFTSV-G-$\Delta$m$^6$A (Fig 7D), along with reduced synthesis of SFTSV RNA (Fig 7E) and G protein (Fig 7F).

The potential role of m$^6$A modification in vaccine design is an emerging topic. The rVSV vaccine platform is widely used to develop various vaccines, including COVID-19 [65], RABV

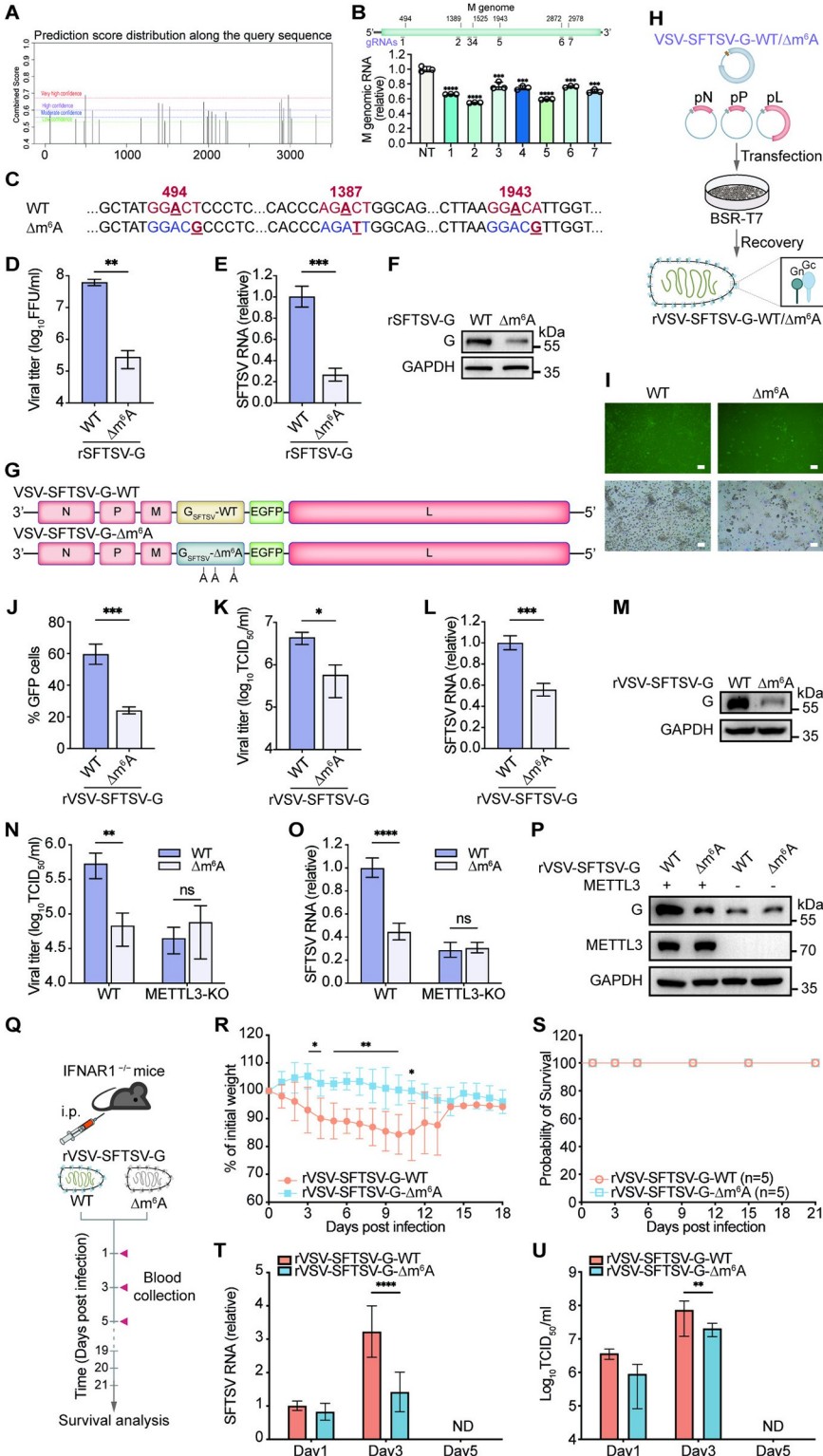

**Fig 7. m⁶A-deficient rVSV-SFTSV-G exhibits replication defects *in vitro* and *in vivo*.** (A-B) Screening of gRNAs targeting SFTSV M genomic RNA with dCas13b-YTHDF2. (A) Prediction score distribution along the sequence of the SFTSV M genome as predicted by SRAMP [60]. (B) Strand-specific RT-qPCR quantifications of knockdown efficiency of dCas13b-YTHDF2 with gRNAs targeting different sites of the SFTSV M genomic RNA to identify regions of the most pronounced effect. (C) Schematic representation of m⁶A sites and introduced synonymous mutations in the

SFTSV G CDS. (D-F) m$^6$A-deficient rSFTSV-G exhibits replication defects in vitro. Huh7 cells were infected with rSFTSV-G-WT/Δm$^6$A at an MOI of 10. (D) Viral titer by focus-forming assay in supernatants of Huh7 cells at 96 hpi. (E) RT-qPCR quantification of intracellular SFTSV RNA using specific primers annealing to the SFTSV G gene. (F) Immunoblot analysis of SFTSV G protein levels in cell lysates harvested at 96 hpi with GAPDH as a loading control. (G) Schematic of the genome structure of VSV-SFTSV-G-WT and VSV-SFTSV-G-Δm$^6$A. (H) Procedure overview for generating rVSV viruses. Recovery of rVSV viruses was achieved by co-transfection of pXN2-SFTSV-G-WT/Δm$^6$A, pN, pP, and pL into BSR-T7 cells. (I-M) m$^6$A-deficient rVSV-SFTSV-G exhibits replication defects *in vitro*. Huh7 cells were infected with rVSV-SFTSV-G-WT/Δm$^6$A at an MOI of 10. (I) Representative images of the GFP expression of rVSV-SFTSV-G-WT/Δm$^6$A at 96 hpi. Scale bar, 100 μm. (J) Percentages of GFP positive cells in Huh7 cells as quantified by ImageJ software. (K) TCID$_{50}$ of supernatants harvested from Huh7 cells at 96 hpi. (L) RT-qPCR quantification of intracellular SFTSV RNA with specific primers targeting G genes, and the data were normalized to ACTB. (M) Immunoblot analysis of G protein levels in extracts of rVSV-SFTSV-G-infected Huh7 cells at 96 hpi. (N) Virus titer by TCID$_{50}$ in supernatants of WT or METTL3 KO cells infected with rVSV-SFTSV-G-WT/Δm$^6$A for 96 h. (O) RT-qPCR of SFTSV RNA or (P) western blot of SFTSV G protein in METTL3 KO or WT Huh7 cells infected with rVSV-SFTSV-G-WT/Δm$^6$A for 96 h. (Q-U) m$^6$A-mutated rVSV-SFTSV-G viruses have defects in replication *in vivo*. (Q) Schematic illustrating the detection of virulence of rVSV-SFTSV-G. 6–8 weeks old IFNAR1$^{-/-}$ C57BL/6 mice were intraperitoneally (i.p.) infected with $2 \times 10^6$ PFU of indicated viruses. Body weight changes (R) and survival rates (S) of mice were monitored continuously till 3 weeks post infection. Blood of mice was collected at day 1, 3, and 5 post infection to determine the levels of SFTSV RNA by using RT-qPCR (T) or sera viral titers by TCID$_{50}$ assay using vero cells (U). Data are representative of three independent experiments and presented as mean ± SD. Statistical significance was determined by one-way ANOVA followed by Dunett's multiple comparisons test (B), student's t test (D, E, J-L and R), or two-way ANOVA followed by Sidak's multiple comparisons test (N-O and T-U). NT, non-targeting guide RNA; ND, not detected; *, $P < 0.05$; **, $P < 0.01$; ***, $P < 0.001$; ****, $P < 0.0001$.

[66] and SFTSV [67]. To construct a recombinant SFTSV vaccine based on the rVSV vector, we cloned either the wild-type or the m$^6$A-mutated SFTSV G gene (GenBank: KX641913.1, Fig 7C) into the pXN2-GFP vector between the *Mlu*I and *Xho*I sites, and generated rVSV-SFTSV-G-WT/Δm$^6$A expressing SFTSV Gn/Gc with an EGFP tag (Fig 7G)., We rescued the replication-competent rVSV-SFTSV-G by co-transfecting BSR-T7 cells with pXN2-SFTSV-G-WT/Δm$^6$A, pN, pP, and pL (Fig 7H) [66, 68]. To investigate the impact of m$^6$A on rVSV-SFTSV-G infection, we infected Huh7 cells with rVSV-SFTSV-G-WT/Δm$^6$A at an MOI of 10. The m$^6$A-mutated rVSV-SFTSV-G showed significantly reduced GFP expression in Huh7 cells at 96 h post-infection compared to WT rVSV-SFTSV-G (Fig 7I and 7J). The Δm$^6$A group had approximately 0.5 logs lower virus titer in the supernatant than the WT group (Fig 7K). The mutated virus also expressed lower levels of SFTSV RNA (Fig 7L) and synthesized less G protein (Fig 7M). These results suggested that m$^6$A-deficient rVSV-SFTSV-G has impaired replication capacity *in vitro*, consistent with the phenotype of those with m$^6$A methyltransferases or YTHDF proteins knockdown. To confirm the replication defects observed in the m$^6$A-deficient rVSV-SFTSV-G, caused by synonymous mutations, are indeed related to the methylation modifications, we challenged WT and METTL3 knockout Huh7 cells with either WT or the m$^6$A-deficient rVSV-SFTSV-G, and showed that the reduction in viral replication was completely abrogated in METTL3-deficient cells (Fig 7N–7P), indicating an m$^6$A-dependent regulation.

We then tested the replication capacity of m$^6$A-deficient rVSV-SFTSV-G *in vivo* (Fig 7Q). IFNAR1$^{-/-}$ C57BL/6 mice were intraperitoneally infected with $2 \times 10^6$ PFU of WT or the m$^6$A-deficient rVSV-SFTSV-G. The body weight decreased in mice during the first 10 days post-infection with wild-type rVSV-SFTSV-G, which gradually recovered afterwards (Fig 7R). As reported previously [67, 69], infection with neither WT nor the m$^6$A-deficient rVSV-SFTSV-G led to mortality in mice (Fig 7S). In keeping with the *in-vitro* data, the m$^6$A-deficient rVSV-SFTSV-G infection resulted in lower viral RNA loads in blood cells and fewer infectious virus particles in the serum than the wild-type virus (Fig 7T and 7U). Thus, the virulence of m$^6$A-deficient rVSV-SFTSV-G was significantly attenuated *in vivo*.

## RNA m$^6$A modification promotes SFTSV infection in *H. longicornis*

m$^6$A modification affects the function of mRNA, long non-coding RNA, or other RNA molecules in diverse organisms, including yeast, plants, drosophila, mammals, and viruses [70]. We next investigated whether m$^6$A regulates SFTSV infection in *H. longicornis*, the primary vector for its transmission. Using BLAST analysis, we identified HPB48_005528 and HPB48_008638 in *H. longicornis* as homologs of m$^6$A methyltransferases METTL3 and METTL14, respectively. Pairwise sequence alignment revealed substantial similarities between HPB48_005528 and METTL3 (50.7% identity and 61.0% similarity), as well as between HPB48_008638 and METTL14 (62.2% identity and 73.9% similarity) (S6A and S6B Fig). Moreover, the structural models of HPB48_005528 and HPB48_008638 overlapped well with human METTL3 (Alpha-Fold DB: Q86U44) and METTL14 (AlphaFold DB: Q9HCE5), with low RMSDs of 1.442 Å and 0.592 Å, respectively (Fig 8A and 8B). Hence, we speculated that HPB48_005528 and HPB48_008638 may act as m$^6$A methyltransferases in *H. longicornis* as well, and referred them as Hlmettl3 and Hlmettl14 in the following analysis, respectively.

We isolated primary cells from tick eggs (Fig 8C) and infected them with SFTSV to examine how these two potential m$^6$A methyltransferases in *H. longicornis* respond to SFTSV infection. SFTSV infection increased Hlmettl3 and Hlmettl14 mRNA levels by about 5-fold and 4-fold, respectively, compared to the mock group (Fig 8D), indicating that these two proteins may be involved in SFTSV infection. We then used dsRNA-mediated RNAi to silence Hlmettl3 (Fig 8E) or Hlmettl14 (Fig 8F) in primary tick cells and further assessed their impact on SFTSV infection. Interfering the transcriptional expression of m$^6$A methyltransferases, especially Hlmettl14, significantly reduced m$^6$A modification of SFTSV L/M/S genomic RNA, as shown by MeRIP combined with strand-specific RT-qPCR (Fig 8G). Downregulation of Hlmettl13 or Hlmettl14 also induced a downward trend in m$^6$A modifications on M antigenomic RNA (Fig 8G). Moreover, silencing Hlmettl14 significantly decreased the expression levels of SFTSV L/M/S genomic RNA and antigenomic RNA as well (Fig 8H). Furthermore, knockdown of m$^6$A methyltransferases in primary cells inhibited SFTSV infection (Fig 8I and 8J), whereas their overexpressions showed an opposite effect (Fig 8K–8N). Together, these results indicated that Hlmettl13 or Hlmettl14 may indeed function as m$^6$A methyltransferases in *H. longicornis*, and that m$^6$A modification regulates the SFTSV lifecycle in ticks by influencing the metabolism of viral RNA.

In addition, we also identified HPB48_005014 as a homolog of m$^6$A binding protein YTHDF in *H. longicornis* by multiple sequence alignment (S7A–S7C Fig), predicted its three-dimensional structure using ColabFold [72] (S7D Fig), and thus referred it as Hlythdf hereafter. Silencing Hlythdf in primary cells decreased the viral load of SFTSV RNA and the production of infectious particles in the supernatant (S7E–S7G Fig), implying that Hlythdf, akin to its m$^6$A binding protein homologs in mammals, fostered SFTSV infection in tick cells as well.

## Discussion

Despite the confirmation of internal modifications in the RNA molecules of various viruses, the types, quantities, and functions of post-transcriptional modifications in SFTSV RNA remain elusive. We here identified various types of post-transcriptional modifications in SFTSV RNAs, including m$^6$A modifications in SFTSV genomic RNA and antigenomic RNA. SFTSV infection changes the expression profiles of several m$^6$A regulators in host cells. Manipulating the expressions or activities of host m$^6$A regulators significantly impacts the replication of SFTSV in cells, suggesting that cellular RNA m$^6$A modifications contribute to SFTSV infection. Furthermore, our data revealed that SFTSV recruits m$^6$A regulators through its NP to modulate the m$^6$A modifications of viral RNA, which promote SFTSV infection by

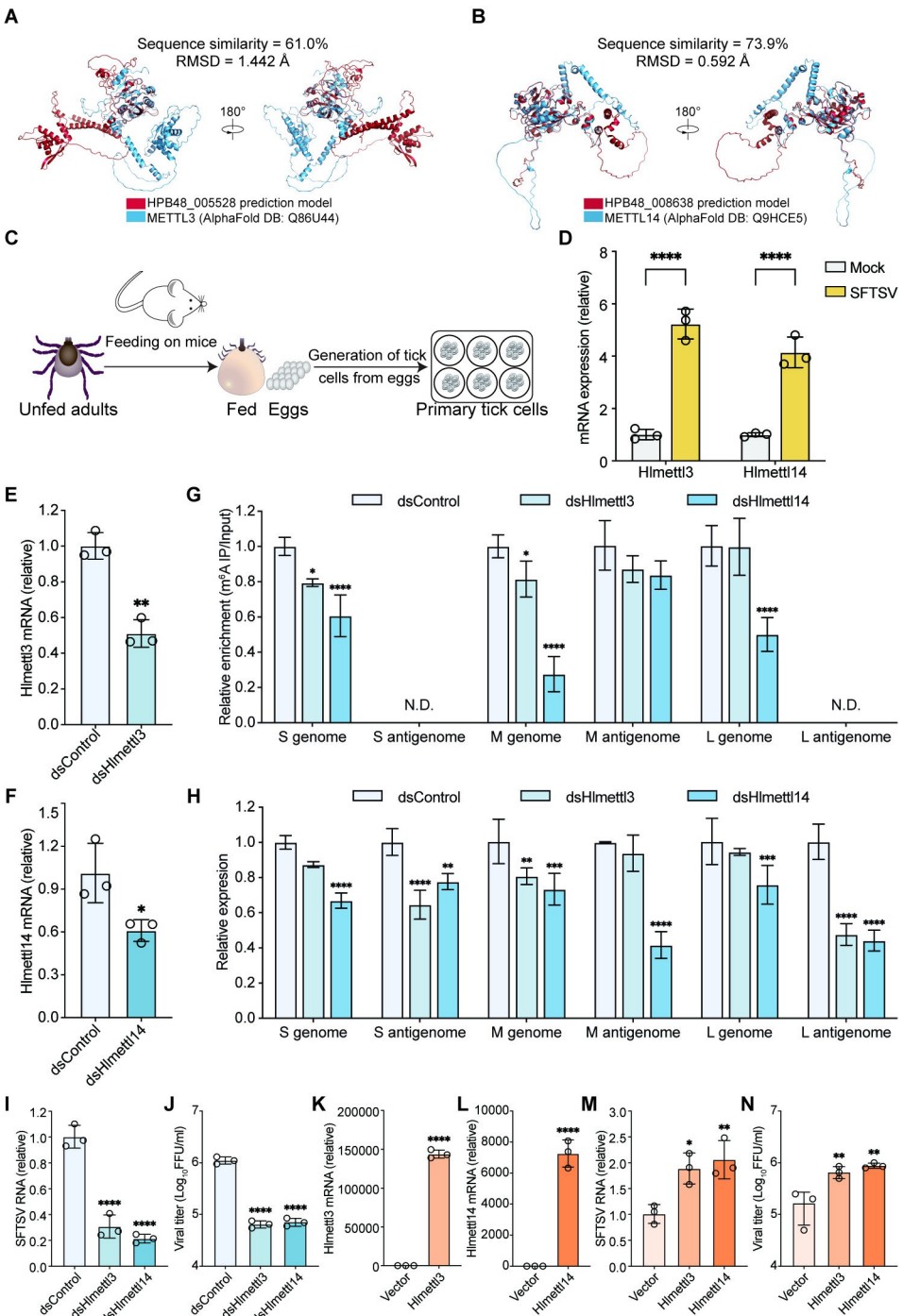

**Fig 8. RNA m⁶A modification promotes SFTSV infection in *H. longicornis*.** (A) Structural superposition of METTL3 structure (blue cartoon, AlphaFold DB: Q86U44 https://alphafold.ebi.ac.uk/entry/Q86U44) and HPB48_005528 prediction model (red cartoon) was performed using PyMOL software[71]. (B) Structural superposition of METTL14 structure (blue cartoon, AlphaFold DB: Q9HCE5 https://alphafold.ebi.ac.uk/entry/Q9HCE5) and HPB48_008638 prediction model (red cartoon) was performed using PyMOL software[71]. (C) Schematic of generation of primary tick cells. (D) RT-qPCR analysis of Hlmettl3 and Hlmettl14 expression in mock- or SFTSV- infected primary tick cells (MOI = 1) at 48 hpi. (E-F) Efficiency of Hlmettl3 (E) and Hlmettl14 (F) knockdown. Primary tick cells were transfected with Hlmettl3 or Hlmettl14 specific dsRNA. Hlmettl3 or Hlmettl14 mRNA were determined by RT-qPCR at 48 h post transfection with Hlactin as the reference gene. (G-H) The m⁶A methyltransferase homologous proteins in *H. longicornis* regulate the m⁶A modification and expression of SFTSV genomic/antigenomic RNA. The m⁶A methyltransferase homologous proteins silenced primary tick cells were infected

with SFTSV at an MOI of 1 for 48 h. m$^6$A modification levels on SFTSV S/M/L genomic or antigenomic RNAs (G) and expressions of SFTSV RNA (H) were assessed using m$^6$A IP followed by strand-specific RT-qPCR. (I) RT-qPCR analysis of viral RNA levels in Hlmettl3 or Hlmettl14 silenced tick cells infected with SFTSV (MOI = 1) at 48 hpi. (J) SFTSV titer in supernatant was measured by focus forming assay. (K-L) Primary tick cells were transfected with plasmids encoding Hlmettl3 or Hlmettl14. mRNA levels of Hlmettl3 (K) or Hlmettl14 (L) were determined by RT-qPCR at 48 h post transfection. (M) qPCR analysis of viral RNA levels in Hlmettl3 or Hlmettl14 overexpressing tick cells infected with SFSV (MOI = 1) at 48 hpi. (N) SFTSV titer in supernatant was measured by focus forming assay. Data are representative of three independent experiments and presented as mean ± SD. Statistical significance was determined by student's t test (D, E, F, K, and L), two-way ANOVA followed by Sidak's multiple comparisons test (G and H), or one-way ANOVA followed by Dunett's multiple comparisons test (I, J, M, and N). N.D., not detected; *, $P < 0.05$; **, $P < 0.01$; ***, $P < 0.001$; ****, $P < 0.0001$.

augmenting the translation efficiency of viral NP mRNA and/or the stability of viral genomic RNA. Notably, the functions of RNA m$^6$A modification seem to be evolutionarily conserved as it facilitates SFTSV infection in tick cells as well (Fig 9).

Increasing evidence has suggested that different RNA modifications may play similar roles in regulating RNA structure, stability, translation, and localization [16]. This functional redundancy implies that various RNA modifications may exert similar functions during viral infection. We therefore detailedly investigated the m$^6$A modification landscape of SFTSV RNAs. In agreement with the MeRIP-seq results on RNA from blood cells of SFTS patients [41], we observed m$^6$A peaks in SFTSV genomic RNA, antigenomic RNA, and mRNAs. Although definitive evidence confirming the presence of antigenomic RNA in SFTSV virions is lacking, our MeRIP-seq analysis revealed that m$^6$A peaks are distributed across all three segments of the SFTSV virion RNA. These findings are consistent with previous observations in other -ssRNA viruses, such as RSV, HMPV, and VSV [27, 28, 48]. However, further experimental validation such as more sensitive detection methods is needed to conclusively determine whether antigenomic RNA is indeed packaged in SFTSV virions. Moreover, our results revealed that distinct m$^6$A methylation patterns in SFTSV RNA between cell types and between cells and virions. The similarity in m$^6$A modifications in S/M genomic RNA suggests selective packaging of modified RNA into virions, potentially aiding in viral replication or immune evasion. In contrast, the variability in m$^6$A peaks in L genomic RNA could be due to differences in packaging strategies, RNA types, or host factors. These results underscore the complexity of m$^6$A modifications in SFTSV RNAs and suggest further investigation into their role in SFTSV pathogenesis.

Given that no literature so far has reported the functional regulation of RNA m$^6$A modification by SFTSV-encoded proteins [56, 73–75], we systemically profiled the expression of host m$^6$A regulators during SFTSV infection, and found that several m$^6$A regulators, including METTL3 and ALKBH5, were upregulated, albeit to different extents, at specific time points post infection. It is thus conceivable that these upregulated host m$^6$A regulators may potentially alter the overall m$^6$A modification of host cells during SFTSV infection. Alterations in m$^6$A levels in viral RNA may directly influence virus infection by regulating the metabolism of viral RNA [35], while m$^6$A modification of host transcripts can indirectly impact the virus lifecycle by regulating host responses [36]. Using RNA editing tools based on the CRISPR-Cas13 system, which enable targeted methylation or demethylation of specific m$^6$A sites on RNA [50, 76, 77], we established the functional correlation between m$^6$A modification in SFTSV genomic RNA and virus infection. Targeted demethylation of SFTSV genomic RNA using dCas13b-ALKBH5/gRNA inhibited SFTSV replication, indicating that m$^6$A modification of SFTSV genomic RNA enhances SFTSV infection. Accordingly, numerous reports have shown that the methyltransferase inhibitor DAA can inhibit the replication/infection of various viruses (RSV, IAV, HIV-1, and SV40) *in vitro* [24,25,29,78–80] and *in vivo* (RSV, Ebola, and

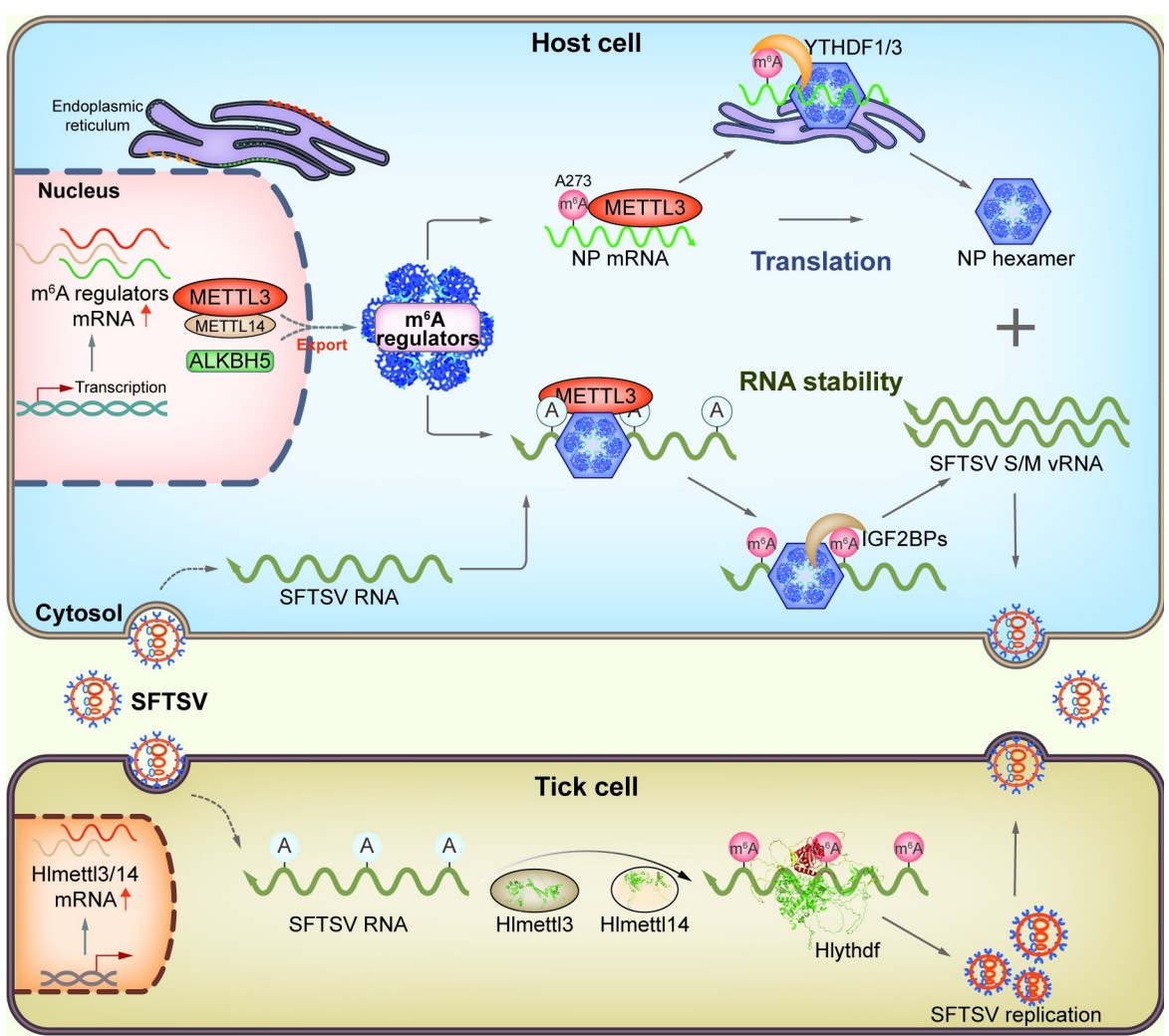

**Fig 9. A schematic illustration of N6-methyladenosine RNA modification promoting SFTSV infection.** During SFTSV infection, several host m⁶A regulators are upregulated. SFTSV recruits m⁶A regulators through the nucleoprotein to modulate the m⁶A modification of viral RNA, which promotes the translation efficiency of viral NP mRNA and the stability of genome RNA, eventually resulting in enhanced viral infection. The function of m⁶A modification in viral RNA is conserved during SFTSV-tick interaction.

etc) [81,82]. Our data here extended these reports by showing that DAA represses SFTSV replications in cells as well, suggesting that drugs selectively inhibiting m⁶A methylation levels have potential therapeutic value against SFTSV infection. Thus, m⁶A may serve as a potential novel target for the development of broad-spectrum antiviral drugs.

Several cytoplasmic replicating viruses, such as HCV, ZIKA, EV71, RSV, HMPV, and SARS-CoV-2, alter the localization of m⁶A regulators upon infecting host cells, thus regulating the m⁶A modification of viral RNA and influencing virus infection [26–28,32,83–85]. As the critical catalytic core of the m⁶A methyltransferase complex, METTL3 normally localizes in the nucleus of cells [86,87], and its abundance increases during the infection processes of various RNA and DNA viruses [33,37,83]. Intriguingly, SFTSV infection significantly increased the expression of METTL3, especially in the cytoplasm of cells where SFTSV replicated. Notably, cytoplasmic METTL3 colocalized with the SFTSV NP protein, which is likely mediated through the direct binding between the ZFD of METTL3 with the C-lobe domain of SFTSV

NP. Importantly, disrupting the interaction between METTL3 and SFTSV NP significantly diminished the promotive effect of METTL3 on SFTSV infection. In addition, it seems that METTL3 is capable of promoting SFTSV infection in an $m^6$A-independent fashion because the truncated METTL3 without the crucial catalytic site (DPPW motif) enhances SFTSV replication as well. These evidences suggest that the domain in METTL3 that interacts with the SFTSV NP could be a novel target for the development of SFTSV-specific antiviral drugs.

Cytoplasmic METTL3 not only enhances the translation of $m^6$A-modified mRNA [20, 58, 88] but also facilitates the translation of some non-$m^6$A-modified transcripts [57]. We noted that the translation efficiency of ectopically expressed NP mRNA was enhanced by METTL3 even with an APPA mutation present, suggesting that the translation-promoting effect is not entirely dependent on its $m^6$A catalytic activity. Cytoplasmic METTL3 may recruit eIF3 to the translation initiation complex to promote the translation of oncogenes such as EGFR and TAZ [58]. Moreover, VSV infection facilitates the translation of $m^6$A-modified IRF3 mRNA by inducing the METTL3-eIF3C interaction in the cytoplasm [89]. Given the presence of both METTL3 and eIF3a in the NP immunoprecipitates as disclosed by our IP-MS data, and that eIF3a, eIF3b, and eIF3c constitute the evolutionarily conserved core subunits of the mammalian translation initiation factor eIF3 [90], we speculate that the recruitment of these eukaryotic translation initiation factors may underlie METTL3-augmented translation of NP mRNA.

We further assessed the impact of each $m^6$A site in the NP CDS on SFTSV NP expression and found that the $m^6$A site at A273 positively regulates NP mRNA translation. Considering the ability of NP to bind YTHDF1/3, and the established roles of these $m^6$A readers in facilitating the translation of $m^6$A-modified mRNA [21,23], we hypothesize that YTHDF1/3 may recognize and bind to the A273 $m^6$A site in NP CDS, thereby participating in the regulation of NP mRNA translation. Nonetheless, further validations are required to confirm whether A273 indeed undergoes $m^6$A methylation in an endogenous context. As the S antigenome sequence also contains the NP mRNA sequence, developing new techniques capable of distinguishing NP mRNA from S cRNA during SFTSV infection would facilitate the investigations on their modifications as well as the functional consequences individually.

Apart from the increased mRNA translation efficiency, our data indicated that $m^6$A modification also enhances the stability of SFTSV RNA, as silencing METTL3 significantly shortened the half-life of both SFTSV S genomic and antigenomic RNAs. NP binds to SFTSV genomic or antigenomic RNA and forms an RNA-protein complex in the lifecycle of SFTSV. In addition, our data showed that SFTSV NP may directly bind to IGF2BP1/2/3 as well, which enhance the stability of $m^6$A-modified RNA [61,91,92]. Therefore, we speculate that the interaction between SFTSV NP and METTL3 may contribute to the proximity of SFTSV genomic RNA or antigenomic RNA to METTL3 in cellular substructures, thereby achieving $m^6$A modification of genomic or antigenomic RNA; subsequently, the bindings of IGF2BP1/2/3 recruited by SFTSV NP to $m^6$A-modified sites catalyzed by METTL3 promote the stability of viral RNA, eventually leading to potentiated virus replication.

We used the reverse genetic system to show that silencing $m^6$A sites *via* synonymous mutations reduced the infectivity and replication of rSFTSV. These findings corroborated the outcomes in our METTL3-knockdown experiments, and further demonstrated the functional impact of these $m^6$A sites on the stability of SFTSV S genomic RNA in endogenous conditions. Indeed, specific $m^6$A site mutations in the genome of RNA viruses such as EV71 and IAV repress the viral replication [25,26]. Recombinant RSV or recombinant HMPV with $m^6$A defects are highly attenuated in cotton rats while retaining high immunogenicity [27,28]. Thus, further investigation on the virulence and immunogenicity of $m^6$A-deficient rSFTSV *in vivo* may lead to the development of attenuated SFTSV live vaccines.

VSV, as a vaccine vector, has great potential to circumvent the strict safety requirements for highly pathogenic viruses like SFTSV [68]. By substituting the VSV glycoprotein (G) gene with the SFTSV Gn/Gc gene that mediates viral entry [93,94], researchers have developed a VSV-based replication-competent recombinant SFTSV vaccine that can be used in a Biosafety Level 2 laboratory [67,69]. Taking advantage of this system, we then determined the roles of $m^6A$ sites in SFTSV M genome on the infection ability of recombinant SFTSV vaccine *in vivo*. Infection of mice with the $m^6A$-mutant rVSV-SFTSV-G resulted in considerably reduced levels of weight loss and serum viral titers as compared to those induced by the WT counterpart, which was attributed, at least partially, to their impaired replication capacity in cells. The impact of these $m^6A$ sites on the immunogenicity of SFTSV as well as the antiviral responses elicited *in vivo* merits further investigations.

We identified a few $m^6A$ sites capable of modulating the stability of SFTSV S/M vRNA. However, the biological functions of other DRACH sites on the viral RNA remain elusive. This may reflect the specificity of $m^6A$ modification, which can be removed or inhibited. The $m^6A$ demethylases ALKBH5 and FTO may catalyze the targeted removal of $m^6A$ modifications from specific transcripts [18,19]. Notably, our IP-MS results posited ALKBH5 as a potential $m^6A$ regulator that interacts with SFTSV NP. Moreover, SFTSV infection altered the expression of METTL3 and ALKBH5, which may subsequently affect the $m^6A$ modification of specific RNAs. In addition, recent research has shown that exon junction complexes (EJCs) act as $m^6A$ "inhibitors" by suppressing the $m^6A$ deposition [95]. EJCs bind to the exon-exon junctions of newly synthesized mRNA, typically 20–24 nucleotides upstream of the upper splice site [96]. However, SFTSV replicates and transcribes in the cytoplasm, and SFTSV-encoded RNA has no splicing capability, as all SFTSV-encoded viral mRNAs (RdRp mRNA, G mRNA, NP mRNA, and NSs mRNA) are single-exon. Therefore, the role of EJCs in regulating the region and transcript selectivity of $m^6A$ modification on SFTSV RNA warrants further investigation.

*H. longicornis* serves as a transmission vector for SFTSV [97], and thus elucidating the role of $m^6A$ in the SFTSV lifecycle in *H. longicornis* is of great importance. The high-precision genome sequence of longhorned ticks enables the study of $m^6A$ regulation in the vector [98]. Our sequence and structural analyses revealed the presence of $m^6A$ methyltransferase homologs in *H. longicornis*, suggesting that $m^6A$ modification is conserved and may occur in ticks. Moreover, we found that $m^6A$ modification promotes SFTSV infection in tick cells. Nevertheless, additional studies, for instance, by knocking down Hlmettl3 or Hlmettl14 *via* microinjection, are needed to further validate the role of $m^6A$ modification in SFTSV infection in tick models. Notably, a recent research showed that the genomic RNA of DENV undergoes $m^6A$ modification in Aag2 cells, and silencing homologous proteins of $m^6A$ methyltransferase/$m^6A$-binding protein YTHDF3 in Aag2 cells reduces DENV infection [99]. Thus, it appears that the $m^6A$ modification as well as its promoting effect on viral infection is a conserved phenotype in ticks. Given that viral infection in vectors typically does not compromise the survival of the vector, further research on how ticks regulate $m^6A$ to limit SFTSV infection to non-pathogenic levels will hold promise for advancing the control of ticks and tick-borne diseases.

## Materials and methods

### Ethics statement

All experiments were performed in accordance with procedures approved by the Ethics Committee of Soochow University and Suzhou Institute of Systems Medicine. Laboratory animal welfare and laboratory licenses are following the National Laboratory Animal Health Guidelines.

## Ticks, Mice, and Animal Housing

The *H. longicornis* strain from Yunnan was maintained as described in our previous publications [100], with a 12-hour light and 12-hour dark cycle at temperatures ranging from 22 to 25˚C, alongside 85% relative humidity.

C57BL/6 mice were obtained from Shanghai Laboratory Animal Center (Shanghai, China). Interferon receptor-deficient (*Ifnar1*$^{-/-}$) C57BL/6 mice were kindly provided by Prof. Chunsheng Dong at Soochow University. Mice were housed in an SPF-grade animal facility at Soochow University. Both male and female mice at the age of 7–9 weeks were used for the experiments.

## Mammalian cells and virus

Vero E6 (kidney epithelial cells from African green monkeys), Huh7, HeLa and HEK293T cells were obtained from ATCC. Wild type (WT) and STAT1 knockout HEK293T cells were a generous gift from Prof. Hui Zheng (Soochow University), and BSR-T7 cells stably expressing T7 RNA polymerase were gifted by Prof. Chunsheng Dong (Soochow University) and treated with 1 mg/ml G418 (Invitrogen). All cells were cultured in Dulbecco′s Modified Eagle Medium (DMEM) with 10% fetal bovine serum (FBS) and 1% penicillin-streptomycin at 37˚C with 5% $CO_2$.

Severe fever with thrombocytopenia syndrome virus (strain SDTA_1) was obtained from China Centre for General Virus Culture Collection and was propagated in Vero cells. All experiments involving the virus were performed in enhanced biosafety level 2 (BSL-2+) facilities, in accordance with the institutional biosafety standard operating procedures.

## Isolation of primary tick cells from eggs of *H. longicornis*

Primary tick cells were prepared according to a previous report with minor modifications [101]. Briefly, when the rectal sac of developing embryos became visible, the female ticks were removed, and the eggs were sequentially treated with 0.1% potassium permanganate, 0.1% benzalkonium chloride, and 70% ethanol followed by rinsing three times with cold PBS. The eggshells were then gently crushed to release the embryonic tissues, which were filtered through a 100 μm filter (Millipore), trypsin-digested, and then centrifuged at 200 g for 5 min. The resulting pellets were washed, cultured in L-15 medium containing 20% FBS and 1% penicillin-streptomycin at 28˚C, and examined daily under a microscope.

## Antibodies

The following primary/secondary antibodies were used in this study: mouse monoclonal antibodies against the m$^6$A (Abcam, ab208577), GAPDH (3F10, Abmart, M20050) or Myc-Tag (9B11, CST 2276S); the mouse IgG1 isotype control (Abcam, ab280974); anti-SFTSV NP serum (prepared in mice) and anti-SFTSV G (ProSci, 6647); rabbit monoclonal antibodies against the DDDDK tag (Abcam, ab205606), Stat1 (D1K9Y, CST, 14994) and METTL3 (Abcam, ab195352); rabbit polyclonal to Lamin B1 (Abcam, ab16048); Alexa Fluor 488-coupled goat anti-rabbit IgG H&L (Abcam, ab150077); Alexa Fluor 647-conjugated goat anti-mouse IgG H&L (Abcam, ab150115); and HRP-linked goat anti-rabbit IgG (CST, 7074) or anti-mouse IgG (Bioworld, 20308020).

## Plasmid construction and site-directed mutagenesis

pcDNA3/Flag-METTL3 and pcDNA3/Flag-METTL14 were gifts from Prof. Chuan He (Addgene plasmid # 53739, # 53740). pFRT/TO/HIS/FLAG/HA-ALKBH5 was a gift from

Prof. Markus Landthaler (Addgene plasmid # 38073). The coding region sequence (CDS) of the full-length METTL3 and ALKBH5 (position 38-VAAAAAAA sequence missing) were amplified from pcDNA3/Flag-METTL3 and pFRT/TO/HIS/FLAG/HA-ALKBH5 by PCR, and then subsequently cloned into pCAGGS-MCS vector between *EcoR*I and *Nhe*I restriction enzyme digestion sites by ClonExpress II One Step Cloning Kit (Vazyme, C112). The CDS of FTO (NM_001080432), YTHDF1 (NM_017798), YTHDF2 (NM_016258), and YTHDF3 (NM_152758) were PCR-amplified from HEK293T cDNA, and then were cloned into pCAGGS-MCS to generate overexpression plasmids. The N-terminal WTAP-binding domain deleted METTL3 truncation (ΔNTD) corresponding to amino acid residues 259–580, and the methyltransferase domain deleted METTL3 truncation (ΔMTD) corresponding to amino acid residues 1–357 were constructed. Moreover, the ZFD domain (aa 259–357)[54, 55] of METTL3 was deleted by PCR amplification of the two flanking regions that were further fused by PCR. METTL3$^{APPA}$ mutation and ALKBH5$^{H204A}$ mutation were introduced by the Quik-Change Lightning Site-Directed Mutagenesis Kit (Agilent, 210518) according to the manufacturer′s instructions.

Cmv d0 dPspCas13b-GGS-Sun and Cmv d0 dPspCas13b-GGS-NYTHDF2(100–200) were gifts from Prof. Bryan Dickinson (Addgene plasmid # 119858, # 119856). dCas13b-ALKBH5 plasmid was constructed by fusing the ALKBH5 CDS to the C-terminus of dPspCas13b delta 984–1090 *via* a flexible (GGS)$_6$ linker. sgRNA-expressing plasmid (pC0043-PspCas13b crRNA backbone) was a gift from Prof. Feng Zhang (Addgene plasmid # 103854). For cloning Cas13 guide RNAs, the top and bottom oligonucleotides were annealed and inserted into the pc0043 vector. For cloning shRNAs, DNA oligonucleotides were annealed and ligated into the *BamH*I/*EcoR*I site of the pGreenPuro lentiviral vector (System Biosciences) according to the manufacturer′s protocol.

Plasmids containing the T7 promoter sequence with SDTA_1 antigenomic RNA segments were created according to the previous description [63,64]. Briefly, viral antigenomic sequence of L (GenBank: KX641909.1), M (GenBank: KX641913.1), and S (GenBank: KX641917.1) segments were synthesized by GENEWIZ and subsequently cloned into a modified pcDNA3.1 (+)/myc-His A plasmid, linearized with *EcoR*I, between a T7 promoter and a hepatitis delta virus ribozyme (HDVR) sequence, resulting in pcDNA3.1(+)-T7-L, pcDNA3.1(+)-T7-M, and pcDNA3.1(+)-T7-S, respectively. The cDNAs were positioned precisely at the sites of transcriptional initiation and ribozyme-mediated cleavage, respectively, to achieve transcription of RNAs that had no additional nucleotides at either terminus by following autocatalytic cleavage by the HDVR. The NP and RdRp ORFs were cloned into the pCAGGS-MCS vector as mentioned before, resulting in pCAGGS-NP-FLAG and pCAGGS-RdRp-FLAG, respectively. The N-terminal arm domain deleted NP truncation (ΔNA) corresponding to amino acid residues 35–245, and the C-lobe domain deleted NP truncation (ΔCL) corresponding to amino acid residues 1–111 were constructed. The N-lobe domain of NP (aa 35–111) [56] was deleted by PCR amplification of the two flanking regions that were further ligated by PCR. For the co-localization and Co-IP assay, the FLAG tag in the pCAGGS-MCS plasmid was replaced by the MYC sequence.

The attenuated VSV backbone pXN2$^{M51R}$-GFP carrying M51R mutation in the M gene of VSV genome, and supporting plasmids pP, pN, and pL were gifted by Prof. Chunsheng Dong (Soochow University) [66,102]. To generate rVSV expressing the SFTSV Gn/Gc protein, the Gn/Gc sequence of SFTSV SDTA_1 strain was PCR-amplified from pcDNA3.1(+)-T7-M and then inserted between *Mlu*I and *Xho*I restriction sites of the pXN2$^{M51R}$-GFP vector. Mutations in the potential m$^6$A sites in SFTSV S genome, NP gene, and G gene were introduced into the pcDNA3.1(+)-T7-S, pCAGGS-NP-FLAG, and pcDNA3.1(+)-T7-M plasmids using Quik-Change site-directed mutagenesis kit.

The *H. longicornis* actin promoter sequence was PCR-amplified from tick cDNA, and then inserted into pAcGFP1-N1 vector between *Hind*III and *BamH*I restriction sites to generate pAcGFP1-Hlactin-pro-N1. The CDS of Hlmettl3, Hlmrttl14, and Hlythdf were PCR-amplified from tick cDNA and were cloned into pAcGFP1-Hlactin-pro-N1 vector between *BamH*I and *Age*I restriction sites to generate pAcGFP1-Hlmettl3, pAcGFP1-Hlmettl14, and pAcGFP1-Hlythdf, respectively. All plasmid sequences were verified by Sanger sequencing, and the sequences of oligonucleotide primers are listed in S1 Table.

## Lentiviral particles, transduction, and transfections

For lentiviral particle production, shRNAs against m6A regulators were co-transfected into HEK293T cells with the lentiviral packaging vector psPAX2 (12260, Addgene) and the envelope vector pMD2.G (12259, Addgene), by using polyethylenimine (PEI) reagent as previously reported [103]. The supernatant containing lentiviral particles was collected 48 hours post-transfection, filtered, and concentrated by centrifugation. Huh7 cells were transduced with the concentrated virus in the presence of 8 μg/ml polybrene, and transduction efficiency was analyzed 3 days later with immunofluorescence microscopy.

Delivery of dsRNAs to primary tick cells was conducted using Lipofectamine 3000 Transfection Reagent (Thermo Scientific, L3000150) according to the manufacturer′s instructions. Media was changed 4 hours post-transfection, and cells were incubated for 24 hours post-transfection prior to each experimental treatment.

## Construction of knockout cell lines by CRISPR-Cas9

To generate endogenous METTL3 knockout cell lines, METTL3 guide RNAs were cloned into the lentiCRISPR-v2 plasmid (Addgene plasmid # 52961) in Huh7 cells using the CRISPR/Cas9 system. The sequences for the guide RNA were as follows: METTL3 guide RNA 1: 5'-CACCGTCTGAACCAACAGTCCACTA-3'; METTL3 guide RNA 2: 5'-CACCGTCAGCA-TAGGTTACAAGAGT-3'. The gRNAs were packaged into lentiviral particles and used to infect Huh7 cells as described earlier. After 72 hours, Huh7 cells were selected using 5 μg/mL puromycin in DMEM supplemented with 10% FBS. After two weeks of culture, individual clones were isolated and expanded under continued puromycin selection. Clonal populations were verified by immunoblotting, and genomic DNA was extracted for PCR and sequencing.

## *In vitro* transcription reaction

To create *in vitro* transcription templates for SFTSV RNA, viral sequences in pcDNA3.1(+)-T7 plasmids were amplified by PCR using corresponding primer pairs containing a T7 promoter sequence (TAATACGACTCACTATAGGG) incorporated at the 5′ end of the products for production of synthetic positive-sense cRNA. The T7 promoter was incorporated at the 3′ end for the production of the synthetic negative-sense vRNA. To prepare dsRNAs for RNA interference in ticks, the T7 promoter sequence was appended to the 5′ ends of the forward and reverse oligonucleotide primers. Resulting PCR products were purified and subjected to *invitro* transcription reaction at 37˚C for 4 h by using the T7 High Efficiency Transcription Kit (Transgene, JT101-01), followed by DNase I digestions of the DNA template at 37˚C for 15 min. The transcript was purified with the EasyPure RNA Purification Kit (Transgene, ER701-01). Concentrations of purified RNA transcripts were determined by using spectrophotometry and the quality of RNAs was examined by denaturing PAGE and ethidium bromide staining.

The molecular copies of synthetic SFTSV RNA were calculated from the total molecular weight and length of each RNA segment. The stocks of *in-vitro* transcribed viral RNAs were

serially diluted to form standard curves used for following RT-qPCR studies. Primers used for *in-vitro* transcription are listed in S2 Table.

## RNA extraction from purified virions and virus-infected cells and m⁶A dot blot assay

After infection with SFTSV, the supernatant and cells were collected separately. SFTSV virions in the supernatant were purified by differential sedimentation through 20–60% sucrose gradients in a Beckman SW32 rotor at 28,000 rpm and 4˚C for 180 min. Viral RNA from purified virions and total RNA from virus-infected cells were extracted by TRIzol reagent. The RNA pellets were resolved in 30 μl RNase-free water after washing with 75% ethanol. DNase I at a final concentration of 1 U/μl was used to degrade cellular DNA at 37˚C for 30 min, followed by inactivation at 65˚C for 10 min. The concentration of RNA was measured using spectrophotometry (BioTek).

The m⁶A dot blot assay was performed as previously described [40]. Briefly, virion RNA or total RNA was denatured by heating at 95˚C for 5 minutes, then immediately chilled on ice. Equal amounts of RNA were loaded onto a Hybond-N+ membrane (GE Healthcare, UK) and UV cross-linked to the nylon membrane. Following UV cross-linking, the membrane was washed with 1× PBST buffer, blocked with 5% non-fat milk, and incubated overnight at 4˚C with primary rabbit anti-m⁶A antibody (Abcam, ab208577). The membrane was then incubated with HRP-conjugated anti-rabbit IgG secondary antibody for 1.5 hours at room temperature with gentle shaking, followed by detection using enhanced chemiluminescence. Equal loading of RNA was verified by 0.02% methylene blue staining.

## Identification of RNA modification by LC-MS/MS

SFTSV RNA extracted from highly purified virions was subjected to reverse transcription using oligo d(T), followed by qPCR for quantification of β-actin. Virion RNA free of contamination of host RNA was used for liquid chromatography-mass spectrometry (LC-MS/MS). Purified RNA was digested and subjected to a quantitative analysis of the modified nucleosides level using LC-MS/MS as previously described [42,104].

## MeRIP-seq and data analysis

High-throughput sequencing of the SFTSV RNA and host cellular RNA was carried out using MeRIP-seq as previously described with minor modifications [105, 106]. Briefly, SFTSV RNA extracted from highly purified virions and total RNA isolated from infected cells were fragmented to 100–200 nucleotides using RNA Fragmentation Reagent (Thermo Fisher, AM8740) by incubating at 95˚C for 5 min. Fragmented RNA was purified using ethanol precipitation with 3 M sodium acetate (pH 5.2) and Glycoblue (Thermo Fisher, AM9515) overnight, followed by the incubation with 5 μg of m⁶A-specific antibody and RNase inhibitor in 500 μl IP buffer (10 mM Tris-HCl pH 7.4, 150 mM NaCl, 0.1% NP-40) at 4˚C for 2 h. 100 μL of pre-washed protein A/G magnetic beads was added to the IP sample and rotated gently for 2 h at 4 ˚C. The beads were washed three times with IP buffer and the bound RNA was recovered through elution with N6-Methyladenosine 5′-monophosphate sodium salt (Sigma, M2780) followed by ethanol precipitation. TruSeq Stranded mRNA Library Prep kit was used to generate cDNA libraries from the input fragmented RNA and IP samples.

MeRIP-seq data were analyzed according to protocols described before [107, 108]. The adapters were trimmed by using TrimGalore (version 0.6.7) with parameters "—phred33 -q 25 -e 0.1—length 15—stringency 3". Reads that cannot be aligned to human rRNA/tRNA sequences by using Bowtie2 (version 2.4.5) [109] with default parameters were mapped to

SFTSV genome and antigenome by using Hisat2 (version 2.2.1) [110]. To identify m$^6$A peaks, the peak calling algorithm MACS2 (version 2.2.7.1) [45] was run with the following parameters:—nomodel—extsize 100 -g 11490. The integrative genomics viewer (IGV) tool (version 2.8.9) [111] was used for the visualization of m$^6$A peaks.

## Strand-specific Reverse transcription and Tagged primer real-time PCR

A 5′ tagged forward primer specific to the SFTSV L/M/S negative strand was used to initiate cDNA synthesis from the vRNA. Similarly, to initiate the cDNA synthesis from cRNA, a 3′ tagged reverse primer was designed to target the SFTSV L/M/S positive strand. The tag consisted of 18–20 nucleotides that showed no homology to SFTSV or *Homo Sapiens* [112]. Reverse transcription was carried out with strand-specific primers (S3 Table) using the Prime-Script RT reagent Kit (Takara, Japan) in accordance with the manufacturer's protocol. Briefly, 500 ng of total RNA from virus-infected cells or 10 pg of *in-vitro* transcribed RNA samples were mixed with 1 µL of the strand-specific RT primer (10 µM) and various amounts of RNase-free water to a final volume of 10 µL. The mixture was incubated at 65˚C for 5 min and then cooled to 4˚C for 2 min. After the addition of the reaction buffer and enzyme mixture, cDNA synthesis was carried out by incubating the sample at 55˚C for 50 min, terminated by heating at 85˚C for 5 min, cooled to 4˚C, and then treated with RNase H at 37˚C for 20 min.

To distinguish between vRNA and cRNA of SFTSV, the real-time PCR was carried out using primer pairs consisting of one primer specific to the tag portion of the RT-tagged primers and the other specific to the appropriate viral sequence (S3 Table). A final volume of 20 µL reaction mixtures consisted of 10 µL of 2 × SYBR Green qPCR Master Mix (Bimake, B21202), 1 µL of each primer (10 µM), 0.4 µL of High ROX Dye (50 ×), 1 µL of cDNA template, and 6.6 µL Nuclease-Free Water. The cycling conditions for the qPCR were 95˚C for 10 min, followed by 40 cycles of 95˚C for 15 sec and 60˚C for 60 sec. A melting curve analysis was then performed (95˚C for 15 s, 60˚C for 1 min, and 95˚C for 15 s) to monitor the specificity of the PCR products. A negative nuclease-free water control was performed on each run. 10-fold serial dilutions of the corresponding synthetic viral RNAs were used to generate a standard curve. The assays were performed in an ABI QuantStudio 1 Real-time PCR detection system and analyzed using the provided software.

## MeRIP-qRT-qPCR

MeRIP-qRT-PCR was performed similarly as MeRIP-seq with some differences. SFTSV virion RNA or total RNA was fragmented, purified by ethanol precipitation, and resuspended in 30 µL RNase-free water. Following MeRIP as described above, the beads were washed with 1 ml IP buffer three times and the bound RNA was purified from the beads using Trizol. 1 µL input and the entire IP fractions were reverse transcribed using the PrimeScript RT reagent Kit (BioRad) and subjected to RT-qPCR or strand-specific RT-qPCR. Primer sequences are supplied in S4 Table. The relative m$^6$A level for each transcript was calculated as the percent of input.

## Quantitative RT-PCR

RNA was reversely transcribed using Prime Script RT Master Kit (Takara, China) using random primers as per the manufacturer's instructions. RT-qPCR was performed in triplicate with 2 × SYBR Green qPCR Master Mix (Bimake, B21202) using the QuantStudio 1 Real-Time PCR System (Thermo Fisher). The level of β-actin mRNA was used for normalization in all qRT-PCR experiments. All primer sequences are listed in S5 Table. The relative expression levels were quantified using the $2^{-\Delta Ct}$ methods.

## RNA immunoprecipitation assay

The RIP assay was performed as described with some modifications [113]. Briefly, HEK293T cells in a 10-cm dish were co-transfected with plasmids of METTL3-FLAG and NP-MYC for 48 h and then subjected to RIP analysis using RiboCluster Profiler RIP-Assay Kit (MBL, RN1005) following manufacturer′s recommendations. The relative enrichment of the transcript amount in the RIP fraction over the amount present in the input sample (RIP/input) was calculated based on normalization to enrichment with the control IgG.

## Cytoplasm and nucleus separation

Cytoplasm and nucleus were separated using NE-PER Nuclear and Cytoplasmic Extraction Reagents (Thermo Scientific, P78833) following the manufacturer's protocol. The separation efficiency was validated by western blot with GAPDH/LaminB1 as markers for the cytosolic/nuclear fractions, respectively.

## Immunoprecipitations (IP) assays

To identify proteins that may interact with SFTSV NP, HEK293T cells seeded in 10 cm dishes were transfected with a total of 10 μg empty plasmid or FLAG-tagged NP plasmid followed by infection with SFTSV at an MOI of 10. At 48 h post-infection, cells were washed twice with cold PBS and then lysed with RIPA buffer (50 mm Tris-HCl pH 7.4, 150 mm NaCl, 1 mM EDTA, 1% NP-40, 5% glycerol) containing protease inhibitors. After incubation for 30 min, the lysates were centrifuged to remove debris. Immunoprecipitations were performed with pre-washed anti-Flag immunomagnetic beads (Bimake, B26102) overnight at 4˚C, after which bound proteins were resolved by SDS-PAGE and analyzed by Coomassie Blue staining. Gel slices were excised and subjected to mass spectrometry analysis.

For exogenous Co-IP assays, HEK293T cells ($2 \times 10^6$) were transfected with a total of 2 μg empty plasmid or indicated expression plasmids. Total proteins were collected 48 hours after transfection. Immunomagnetic beads with pre-coupled antibodies were mixed with supernatants of cell lysates overnight at 4˚C. Immunoprecipitated proteins were subjected to western blot analysis.

For endogenous Co-IP assays, Huh7 cells were infected with SFTSV at an MOI of 1. Cell lysate harvested at 48 hours post infection was subjected to primary antibodies or corresponding control IgG and rotated for 3 h at 4˚C, and then incubated with protein A/G agarose overnight at 4˚C. The immunoprecipitations were washed with lysis buffer five times before the immunocomplex was eluted.

## Western blotting

Briefly, protein samples were extracted using cell lysis buffer (Beyotime, P0013) on ice and then centrifuged at 4˚C. Subsequently, protein concentrations were determined using Pierce BCA Protein Assay Reagent (Thermo Fisher, 23228). Total proteins were separated by gradient sodium dodecyl sulfate-polyacrylamide gel electrophoresis (SDS-PAGE), transferred onto polyvinylidene fluoride (PVDF) membranes, blocked with 5% non-fat milk, and incubated with primary antibodies overnight at 4˚C at the dilution as per the manufacturer′s protocols. After being washed 3 times, the membrane was incubated with horseradish peroxidase (HRP) conjugated secondary antibodies for 1 h at room temperature. Protein bands were visualized by an enhanced chemiluminescence reaction using a Tanon imaging system (Tanon, China).

## Immunofluorescence

HeLa cells in 12-mm coverslips were transfected with the indicated plasmids, and then mock infected or infected with SFTSV at an MOI of 10. 48 h post-infection, cells were fixed with 4% paraformaldehyde, permeabilized with 0.1% Triton X-100, blocked with 5% FBS, and incubated with primary antibodies. Alexa Fluor 488/647-coupled antibodies and DAPI were used to acquire images with a Nikon Eclipse Ti-S inverted microscope.

## MTT Assay

To study the effects of DAA on cell viability, Huh7 cells were seeded in 96-well plates at 10,000–20,000 cells/well in triplicate. Once cells were attached, the media were replaced with 100 μl/well of 2% FBS/DMEM containing various concentrations of DAA. After two days' culture in DAA, an MTT assay (Solarbio, M1020) was used to assess cell viabilities following the manufacturer's instructions.

## Focus-forming Assay (FFA)

Vero cells seeded in 24-well plates with 80% confluence were incubated with ten-fold serially diluted SFTSV in DMEM at 37°C for 2 h, followed by the addition of an overlay consisting of 2% FBS/DMEM supplemented with 1% (w/v) methylcellulose (Sigma-Aldrich). After incubation at 37°C for 7 days, the overlay was removed, and the cells were fixed with 4% formaldehyde and then permeabilized with 0.1% Triton X-100. The fixed/permeabilized cells were probed with a 1:200 dilution of anti-NP serum followed by a 1:1,000 dilution of horseradish peroxidase (HRP)-labeled anti-mouse IgG (Bioworld). Visualization of viral foci was accomplished by using KPL TrueBlue Peroxidase Substrate (SeraCare). Viral titers were calculated as focus forming units (FFU) per milliliter (mL) as previously described [114].

## TCID$_{50}$

Triplicate samples of uniform layers of Vero E6 cells in 96-well plates were incubated with 100 μl virus diluted in a series of tenfold. Seven days after inoculation, the cell cytopathic effects were scored and the viral titer was calculated by the method of the Reed-Muench [115].

## Luciferase reporter assay

To screen gRNAs showing the largest response with the dCas13b-YTHDF2 construct, HEK293T cells in 24-well plates were transfected with 500 ng dCas13b-YTHDF2, 300 ng gRNA, and 30 ng of dual-luciferase reporter per well. After 24 hours, cells in each well were digested with trypsin, washed with PBS, and reseeded into a 96-well plate and a 12-well plate. 12 hours after reseeding, cells in the 96-well plate were assayed by Dual-Glo Luciferase Assay Systems (Promega, E2920), while cells in the 24-well plate were processed to extract total RNA (DNase I digested) followed by RT-PCR quantification. For both RNA and protein measurements, the Fluc signal was normalized by the Rluc dosing control.

The NP ORFs containing the wild-type m$^6$A motifs, as well as mutant motifs (see Fig 5J), were inserted into the site right before the stop codon of the firefly luciferase reporter gene of the pmirGLO vector. The A or C within the m$^6$A consensus motif was mutated to a T or G in these sites without changing the encoded amino acid. To evaluate the translation efficiency of SFTSV NP, 50 ng pmirGLO-NP-CDS and 450 ng expression plasmids (pCAGGS-MCS, pCAGGS-METTL3) were co-transfected into HEK293T cells in a 6-well plate at 60–80% confluency. After 24 h, cells in each well were digested with trypsin, washed with PBS, and reseeded into a 96-well plate (1:20) and a 24-well plate (1:2). 12 hours after reseeding, cells in

the 96-well plate were assayed by Dual-Glo Luciferase Assay Systems (Promega, E2920). Firefly luciferase (F-luc) activity was normalized by Renilla luciferase (R-luc) to evaluate the translation of the reporter. Samples in the 24-well plate were processed to extract total RNA (Dnase I digested) followed by RT-PCR quantification. The amount of F-luc mRNA was also normalized by that of R-luc mRNA. The translation efficiency of SFTSV NP is defined as the quotient of reporter protein production (F-luc/R-luc) divided by mRNA abundance [21].

To determine whether METTL3-induced expression of SFTSV NP is dependent on $m^6A$ modification, we performed dual-luciferase reporter and mutagenesis assays with pmir-GLO-NP-CDS-WT (wild-type CDS of SFTSV NP) and pmirGLO-NP-CDS-mut1/2/3 (mutant CDS of SFTSV NP, see Fig 5J). All the plasmids were transfected into METTL3-knockdown HEK293T cells. The relative luciferase activities were assessed with Dual-Glo Luciferase Assay Systems (Promega, E2920) at 48 hours. Each group was repeated in triplicate.

## RNA stability assay

To evaluate the stability of SFTSV RNA, the genomic or antigenomic S segment was inserted behind the F-luc coding region of pmirGLO plasmids to generate pmirGLO-S-genome and pmirGLO-S-antigenome, respectively. 200 ng reporter plasmids (pmirGLO-S-genome or pmirGLO-S-antigenome) and 1800 ng effecter plasmids (shCtrl or shMETTL3) were transfected into Huh7 cells in a well of 6-well plate at 60–80% confluency. After 24 h, each well was trypsin-digested, washed with PBS, and re-seeded into five wells of a 24-well plate. 24 h after reseeding, Actinomycin D (Selleck, S8964) was added to shCtrl and shMETTL3 cells at 5 μg/ml to assess RNA stability. After incubation for indicated times, the cells were collected and RNA samples were extracted for reverse transcription and qPCR. The F-luc mRNA degradation rate was estimated using ln2/slope and R-luc mRNA was used for normalization, according to published protocols [17,59,116].

## Recovery of rSFTSVs or rVSVs from the full-length cDNA clones

rSFTSVs were rescued from the full-length cDNA of the SFTSV (SDTA_1 strain) according to previously published reports [63,64]. Briefly, BSR-T7 cells cultured in 10 cm dishes with 75% confluence were transfected with 2.5 μg of pCAGGS-NP-FLAG, 2.5 μg of pCAGGS-RdRp-FLAG, 5 μg of pcDNA3.1(+)-T7-L, 5 μg of pcDNA3.1(+)-T7-M, and 5 μg of pcDNA3.1(+)-T7-S using Lipofectamine 3000 Reagent (Thermo Fisher). At 5 days post-transfection, cells were subjected to three freeze-thaw cycles and centrifuged at 4000 × g for 10 min. The supernatant was used to infect fresh Vero cells for 7 days before collection. The successful recovery of the rSFTSVs was confirmed by FFA, followed by RT-PCR and sequencing.

Recovery of rVSVs was conducted as described previously by co-transfection 10 μg of pXN2-SFTSV-G and three helper plasmids pN (5 μg), pP (4 μg), and pL (2μg) into BSR-T7 cells cultured in a 10 cm dish [66–69]. Upon detection of the cytopathogenic effect, the supernatants containing pseudotype viruses were collected and further expanded in Vero cells for one passage to generate a stock [67]. The successful recovery of the rVSVs was confirmed by the presence of green fluorescent cells, followed by RT-PCR and sequencing.

## Replication of rVSV-SFTSV-G in mice

The 6–8 weeks old male and female *Ifnar1*$^{-/-}$ mice were randomly divided into two groups (n = 5/group) and infected intraperitoneally with $5 \times 10^7$ PFU of WT or $m^6A$ mutated rVSV-SFTSV-G. Body weights and clinical symptoms were monitored continuously for 21 days after infection. Whole blood samples were collected at 1, 3, and 5 days post infection for viremia analysis.

## Sequence and structural alignment

The protein sequences of human m$^6$A regulators were downloaded from Uniprot [117]. NCBI Basic Local Alignment Search Tool (BLAST) (blastp) was used to identify *H. longicornis* protein sequences homologous to human m$^6$A regulators [118]. Protein pairwise sequence alignment (PSA) and multiple sequence alignment (MSA) were performed using the Clustal Omega web server of the European Bioinformatics Institute (EMBL-EBI) [119]. Espript 3 software [120] was used to represent the PSA and MSA.

The protein structure predictions of human m$^6$A regulators were downloaded from the AlphaFold protein structure database [121,122]. ColabFold was employed to build protein structure predictions of *H. longicornis* homologs of human m$^6$A regulators, followed by a Foldseek quest to identify structural similarities within the PDB and AlphaFold databases [72]. The structures of m$^6$A regulators homologs were superimposed aligned in PyMOL using the super function [71].

## Statistical analysis

Statistical analyses were performed using GraphPad Prism 9 software. The results are expressed as mean ± standard deviation (SD) from three or more independent experiments unless otherwise noted. 2-sided unpaired Student's t-test was used for two groups comparison, one-way analysis of variance (ANOVA) with post Tukey′s multiple comparisons test, and two-way ANOVA with post Sidak′s multiple comparisons test were used for multi-group comparison. $P < 0.05$ was statistical significance.

## Supporting information

**S1 Fig. The m$^6$A methylome of SFTSV plus-sense RNA.** (A) Schematic diagram of strand-specific RT-qPCR. The strand-specific tagged reverse transcription primer (Tagged-RT primer) carrying a non-viral tag at the 5' end is reverse complementary to the 3'-UTR region in the genome or antigenome. The qPCR uses a non-viral tag as forward primers, and strand-specific reverse primers are designed on the genome or antigenome. Using the plasmid encoding the SFTSV S genome as a template, the RNA of the S genome or antigenome was *in-vitro* transcribed to make the corresponding standard curve. (B) Stranded-specific RT-qPCR detection of the copy numbers of SFTSV S genome/antigenome. The total RNA isolated from Huh7 cells infected with SFTSV for 48 h was used as a positive control, and ddH$_2$O was used as a negative control (NC). (C) RT-qPCR detection of ACTB mRNA with total RNA from (B) as a positive control and ddH$_2$O as the NC. ACTB mRNA was not detectable in cDNA reversely transcribed from virion RNA using Oligo d(T). Data are representative of three independent experiments and presented as mean ± SD. (D-F) The distribution of m$^6$A peaks in the S, M and L segments of SFTSV plus-sense RNA (+ssRNA). MeRIP-seq mapping of SFTSV RNAs derived from either purified virions (D) or SFTSV-infected Huh7 (E) or HeLa (F) cells. Antigenome/+ssRNA: The baseline signal of Input was displayed as blue, and the m$^6$A IP signal was displayed as red. The gray rectangle indicated the m$^6$A peaks recognized by MACS2 [45]. (TIF)

**S2 Fig. The knockdown efficiency of shRNAs for m$^6$A methyltransferases, demethylases and readers.** HEK293T cells were co-transfected FLAG-tagged m$^6$A methyltransferases (METTL3 in A, and METTL14 in B), demethylases (ALKBH5 in C, and FTO in D) or readers (YTHDF1/2/3 in E-G, respectively) together with a control (shCtrl) or two different shRNAs targeting the overexpressed protein. mRNA and protein levels of each overexpressed molecule were measured by RT-qPCR with ACTB as the reference gene and by WB with GAPDH as the

loading control, respectively. Data are representative of three independent experiments and presented as mean ± SD. Statistical significance was determined by one-way ANOVA followed by Dunett's multiple comparisons test. **, $P < 0.01$; ***, $P < 0.001$; ****, $P < 0.0001$.
(TIF)

**S3 Fig. m⁶A readers YTHDF1/2/3 promote SFTSV infection.** (A) Knockdown of YTHDF1/2/3 inhibits SFTSV infection. Huh7, HeLa or HEK293T cells were infected with lentiviral particles containing either control (shCtrl) or m⁶A readers-specific (shYTHDF1/2/3) shRNAs and selected with puromycin. (B) Overexpression of YTHDF1/2/3 promotes SFTSV infection. Huh7, HeLa, or HEK293T cells were transfected with plasmids encoding YTHDF1/2/3. After 24 h, cells were infected with SFTSV (MOI = 1) for 48 h. Relative cellular SFTSV RNA were quantified by RT-qPCR at 48 hpi with ACTB as the reference gene, while the released SFTSV RNA isolated from equal volumes of supernatants were quantified by RT-qPCR and normalized to the control group. Total cell extracts harvested at 48 hpi were subjected to western blot using antibody against SFTSV NP protein with GAPDH as a loading control. Data are representative of three independent experiments and presented as mean ± SD. Statistical significance was determined by one-way ANOVA followed by Dunett's multiple comparisons test. *, $P < 0.05$; **, $P < 0.01$; ***, $P < 0.001$; ****, $P < 0.0001$.
(TIF)

**S4 Fig. Targeted demethylation of SFTSV genomic RNA inhibits SFTSV infection.** (A) Schematic of dCas13-ALKBH5 targeting assay. SFTSV-infected Huh7 cells were transfected with plasmids encoding dCas13b-ALKBH5 fusion protein and dCas13 guide RNAs targeting different SFTSV genomic RNAs. (B-D) m⁶A modification levels on S (B), M (C), and L (D) genome RNAs were determined by m⁶A-IP followed by RT-qPCR analysis, with HPRT1/DICER as negative/positive controls, respectively. (E) Expressions of SFTSV RNAs in SFTSV-infected Huh7 cells, which had been transfected with dCas13b-ALKBH5 and different gRNAs to target different sites of the viral genome RNA, was quantified by RT-qPCR. (F) The methylation inhibitor DAA restricts SFTSV infection. RT-qPCR analysis of SFTSV RNA expression or western blot analysis showing NP protein levels in DAA-treated and SFTSV-infected (MOI = 1) Huh7 cells at 48 hpi. (G) Cell viabilities of Huh7 cells after 48 h DAA treatment were assessed by the MTT assay. Data are representative of three independent experiments and presented as mean ± SD. Statistical significance was determined by two-way ANOVA followed by Sidak's multiple comparisons test (B-D), or one-way ANOVA followed by Dunett's multiple comparisons test (E and F). **, $P < 0.01$; ***, $P < 0.001$; ****, $P < 0.0001$.
(TIF)

**S5 Fig. SFTSV NP interacts with several m⁶A readers.** (A) Potential NP-interacting m⁶A regulators identified by IP-MS analysis. (B) Co-IP of METTL3 with several SFTSV proteins. HEK293T cells were co-transfected with Myc-tagged METTL3 plasmid and each Flag-tagged NP/NSs/G plasmid. At 36 h post transfection, Co-IP was performed by incubating the cell lysates with the anti-Flag antibody-coupled magnetic beads overnight, and the resulting immunoprecipitates were analyzed by western blotting with indicated antibodies. (C) Co-IP assay of the endogenous METTL3 and NP in SFTSV-infected Huh7 cells using control IgG, anti-METTL3 or anti-NP antibodies, followed by western blotting. (D) Representative confocal microscopy images of mock- or SFTSV-infected HeLa cells (48 hpi, MOI = 1) expressing MYC-tagged NP and FLAG-tagged METTL3 proteins. Nuclei (blue) were strained with DAPI; METTL3 (green) and NP (red) were labeled using anti-FLAG and anti-MYC antibodies, respectively. Scale bar, 10 μm. (E) Nuclear and cytosolic fractions of mock- or SFTSV-infected HEK293T cells (48 hpi, MOI = 1) expressing FLAG-tagged METTL3 were separated and

probed by immunoblotting. GAPDH/LAMIN B1 were used as cytoplasmic/nuclear markers, respectively. (F-H) Representative confocal microscopy images of mock- or SFTSV-infected (48 hpi, MOI = 1) HeLa cells expressing MYC-tagged NP and FLAG-tagged YTHDF1/2/3. The nuclei (blue), YTHDF1/2/3 (green), and NP (red) were detected as in panel D. Scale bar, 10 μm. (I-K) Co-IP of SFTSV NP with YTHDF1/2/3 in HEK293T cells co-transfected with FLAG-tagged YTHDF1/2/3 and MYC-tagged NP plasmids for 36 h. Co-IP and western blot were performed as described in panel B. (L-N) Representative confocal microscopy images of mock- or SFTSV-infected HeLa cells expressing MYC-tagged NP and FLAG-tagged IGF2BP1/2/3, detected as in panel D. Scale bar, 10 μm. (O-Q) Co-IP of SFTSV NP with IGF2BP1/2/3 in HEK293T cells co-transfected with FLAG-tagged IGF2BP1/2/3 and MYC-tagged NP plasmids for 36 h. Co-IP and western blot were performed as described in panel B.
(TIF)

**S6 Fig. Sequence alignment of *H. longicornis* m$^6$A writers.** (A) Pairwise sequence alignment between *H. Iongicornis* HPB48_005528 and *H. sapiens* METTL3 protein. The DPPW motif is represented by red stars. (B) Pairwise sequence alignment between *H. Iongicornis* HPB48_008638 and *H. sapiens* METTL14 protein. The METTL14 R298 residue is represented by a red star. The alignment was performed using the EMBOSS Water web server [119] and visualized by ESPript 3 software [120]. The red highlighted residues are identical, while residues highlighted in yellow are conserved between the two proteins. Secondary structures are represented at the top of the PSA, and the consensus sequence is shown at the bottom.
(TIF)

**S7 Fig. The YTHDF homologous protein in *H. longicornis* promotes SFTSV infection.** (A-B) Multiple sequence alignment for *H. longicornis* HPB48_005014 and *H. sapiens* YTHDF proteins. The alignment was conducted using the Clustal Omega web server [119], and visualized with ESPript 3 software [120]. Identical/conserved residues among the four proteins are highlighted in red/yellow, respectively, with the consensus sequence at the bottom. (A) Alignment of HPB48_005014 at the region in aa 544–829. (B) Alignment of HPB48_005014 at the region in aa 831–1032. (C) Domain architectures of HPB48_005014 and YTHDF proteins. (D) A predicted structure of HPB48_005014 protein generated by ColabFold [72] and visualized using the PyMOL software [71]. Regions in aa 831–1032, aa 544–829, and the remainder are colored with red, yellow, and green, respectively. (E) Efficiency of Hlythdf knockdown. Primary tick cells were transfected with Hlythdf specific (dsHlythdf) or control (dsControl) dsRNA, and relative expressions of Hlythdf mRNA were determined by RT-qPCR with Hlactin as the reference gene. (F-G) The Hlythdf knockdown cells were infected with SFTSV at an MOI of 1. (F) Relative SFTSV RNA levels at 48 hpi were quantified by RT-qPCR with Hlactin as the reference gene. (G) Viral titers in supernatants were assessed by FFA at 48 hpi. Data are representative of three independent experiments and presented as mean ± SD. Statistical significance was determined by student's t test. **, $P < 0.01$; ***, $P < 0.001$.
(TIF)

**S1 Table. Oligonucleotide sequences used for plasmids construction.**
(DOCX)

**S2 Table. Primers for preparation of DNA templates for *in vitro* transcription.**
(DOCX)

**S3 Table. Primer sets for strand-specific reverse transcription qPCR.**
(DOCX)

**S4 Table. Oligonucleotide sequences of primers used in MeRIP-qPCR.**
(DOCX)

**S5 Table. Oligonucleotide sequences of primers for RT-qPCR.**
(DOCX)

## Acknowledgments

The authors acknowledge Prof. Hui Zheng (Soochow University) for sharing WT and STAT1 knockout cells. We thank Prof. Chunsheng Dong (Soochow University) for kindly sharing BSR-T7 cells and plasmids for the recovery of rVSV-SFTSV-G.

## Author Contributions

**Conceptualization:** Feng Ma, Qihan Wu, Jianfeng Dai.

**Data curation:** Zhiqiang Chen, Jinyu Zhang, Feng Ma, Qihan Wu, Jianfeng Dai.

**Formal analysis:** Zhiqiang Chen, Jinyu Zhang, Jun Wang.

**Funding acquisition:** Feng Ma, Jianfeng Dai.

**Investigation:** Zhiqiang Chen, Jinyu Zhang, Jun Wang, Hao Tong, Feng Ma, Qihan Wu, Jianfeng Dai.

**Methodology:** Zhiqiang Chen, Jinyu Zhang, Hao Tong, Wen Pan.

**Resources:** Jianfeng Dai.

**Supervision:** Jianfeng Dai.

**Visualization:** Zhiqiang Chen.

**Writing – original draft:** Zhiqiang Chen, Jun Wang, Jianfeng Dai.

**Writing – review & editing:** Zhiqiang Chen, Jinyu Zhang, Jun Wang, Hao Tong, Wen Pan, Feng Ma, Qihan Wu.

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
