## [Decision Letter · Decision Letter 0]

23 Jul 2024

Dear Dr. Dai,

Thank you very much for submitting your manuscript "N6-methyladenosine RNA modification promotes Severe Fever with Thrombocytopenia Syndrome Virus infection" for consideration at PLOS Pathogens. As with all papers reviewed by the journal, your manuscript was reviewed by members of the editorial board and by several independent reviewers. In light of the reviews (below this email), we would like to invite the resubmission of a significantly-revised version that takes into account the reviewers' comments. All three reviewers recognized your attention to an interesting and important topic and the comprehensive nature of your study. However, there were significant concerns noted and additional experiments suggested by reviewers that would improve the impact and significance of your work. These points must be addressed in a revised version of your manuscript.

We cannot make any decision about publication until we have seen the revised manuscript and your response to the reviewers' comments. Your revised manuscript is also likely to be sent to reviewers for further evaluation.

Sincerely,

Holly Ramage

Guest Editor

PLOS Pathogens

Matthias Schnell

Section Editor

PLOS Pathogens

Michael Malim

Editor-in-Chief

PLOS Pathogens

orcid.org/0000-0002-7699-2064

Reviewer's Responses to Questions

**Part I - Summary**

Reviewer #1: Severe Fever with Thrombocytopenia Syndrome Virus (SFTSV), or Dabie bandavirus (DBV), is a tick-transmitted (-) ssRNA virus that causes severe fever with thrombocytopenia syndrome with a mortality rate up to 12%. N6-methyladenosine (m6A) is a prevalent epigenetic modification of cellular and viral RNA, regulating gene expression. However, the role of m6A modification during SFTSV infection remains elusive. In this manuscript, Chen et al. discovered the m6A modification to SFTSV genome, which facilitates viral replication. Approximately 0.32% of A bases in SFTSV RNA were m6A methylated, similar to the level in host mRNA. siRNA-mediated knockdown oh m6A writer, METTL3 or METTL14, reduced, while overexpression promoted viral replication in multiple cell lines. Knockdown of a m6A eraser, ALKBH5, increased, while overexpression reduced viral replication. iTRAQ-based quantitative proteomic analysis revealed that SFTSV infection upregulated these m6A writers and erasers. Mechanistically, SFTSV nucleoprotein (NP) interacted with METTL3, and m6A modification of viral RNA genome enhanced viral RNA stability and translation efficiency. The authors identified the m6A targets in the genome of SFTSV and made mutant viruses (VSV-pseudotyped). The mutant viruses were less infective and pathogenic in mice and cell cultures, when compared to their parental strain. Interestingly, the m6A modification to SFTSV genome, and its role in viral replication were recapitulated in tick cells. This is a very comprehensive study with rigorous approaches and analyses, novel and solid results, reasonable conclusions. The manuscript is well organized and clearly written.

Reviewer #2: The tick-borne Thrombocytopenia Syndrome Virus (SFTSV) can cause sever fever and serious health concerns to human health. This manuscript used multiple research techniques to identify that N6-methyladenosine (m6A) RNA modification promotes SFTSV infection in human cells, mice, and ticks. The authors provided several lines of evidence and elegantly supported their findings. As the reviewers, I only have some minor concerns with the style of writing and request a minor revision.

Reviewer #3: N6-methyladenosine (m6A) is a very common post-transcriptional modification found in both host and viral RNAs. It has been shown to regulate various viral infections in addition to the roles in modulation of cellular biological processes. Previous studies have reported m6A modification in SFTSV genomic RNAs and shown the potential involvement of multiple related regulatory proteins during the SFTSV infection. However, detailed functional implications need further depiction. In this paper by Chen et al., several possible proviral roles of m6A modification in NP/S and GP/M RNAs are proposed. Additionally, positive regulation of the viral infection potentially by m6A is also implicated in H. longicornis tick cells. The manuscript is interesting and clearly written on the whole. However, there are several significant concerns (especially regarding the inconsistency of some major results and lacking of mechanism elucidation as well as an appropriate in vivo evaluation) that require to be comprehensively addressed.

**Part II – Major Issues: Key Experiments Required for Acceptance**

Reviewer #1: The authors have presented comprehensive and convincing results to support their conclusions.

Reviewer #2: NA

Reviewer #3: 1) SFTSV is a negative-sense “single-stranded” RNA virus, which means that the virions contain negative-sense genomic RNA segments. How are m6A modifications in antigenomic RNAs abundantly detected and quantified in the virion particles? If that is the case, further investigations and additional data and discussions are required here.

Moreover, I suggest integrating the results of intracellular m6A methylome sequencing analysis into the main text rather than placing them in supplementary materials. It is unclear whether packaging integration of antisense RNA can occur (previous evidence does not support it for SFTSV), what the efficiency is, and whether it is uniform. Additionally, please carefully compare, analyze, and discuss the similarities and differences between the sequencing results in this article and those in the previous paper, and provide corresponding explanations.

2) Fig 2 illustrates the positive role of m6A modification in SFTSV infection, which is a major result of the paper. However, the data presented here are not sufficiently reliable or comprehensive.

For instance, only one shRNA was used in the knockdown experiment for functional validation, further verification of the actual knockdown effect by WB is lacking, and many results show a very weak impact (especially in the quantification of RNA copies, difference of only ~0.1 lg was observed in several comparisons), among other issues.

Additionally, even under the same experimental conditions within the same cells, if we only look at the control groups, the quantitative differences in RNA copy numbers are extremely large and severely inconsistent. For example, in Figures 2A and 2C, the shCtrl groups show lg RNA copies: in Huh7 cells, 6.7 vs 5.2; in HeLa cells, 3.5 VS 5; in HEK293T, 3.5 VS 4.8. Furthermore, in Figures 2B and 2D, the Vector groups demonstrate: in Huh7 cells, 4.5 lg vs 6 lg; in HeLa cells, 3.9 lg VS 6.9 lg (meaning 1000-fold difference in virus yields under the same experimental settings); in HEK293T, 3.5 lg VS 5.2 lg, etc.

The results in Fig 2H and 2I ruling out the involvement of the interferon system are also a bit weak. Why were cells with a heterozygous STAT1 deficiency (STAT1+/-) used, instead of cells with a homozygous STAT1 defect (STAT1-/-)?

These severely affect the quality and soundness of the paper and may need to be thoroughly checked and reinforced.

3) Figure 4 shows a potential interaction between A and B, which is considered a mechanism analysis. However, the data presented here are also insufficient and uncertain at the current state.

For instance, both METTL3 and NP are RNA-binding proteins. Is the interaction between METTL3 and NP mediated by RNA in a non-specific manner? Endogenous METTL3 interaction/colocalization with NP also need to be tested in the context of SFTSV infection. Also, the biological significance of this interaction still requires further experimental validation. Does METTL3 interact with other viral proteins? That is, at least, a control with other viral proteins is required.

Further, in Figure 4H, the authors claim that the ZFD domain is crucial for the interaction between NP and METTL3, but the precipitation of the delZFD (when used as bait protein) itself is significantly lower than other truncations or the wild-type protein. This result needs to be repeated and verified. Moreover, from the results in Figure 4K, it can be seen that even such a large deletion of the whole MTD domain (which totally deprives of methyltransferase activity) still retains the pro-viral activity. This activity, however, is only slightly lower than that of the full-length protein, despite its expression being significantly less. These suggest that even with a complete loss of the enzymatic activity, there is no significant reduction in its pro-viral function. Thus, the extent to which the observed pro-viral activity is related to its methyltransferase activity remains uncertain.

4) Related to some of the concerns mentioned above, cell lines with m6A writers (especially METTL3) knockout could be very useful for verifying many results. The KO cell lines could not only be used in the functional studies of Figure 2A but also in the research on drug activity (as shown in Figures 2F and 2G) or site-directed RNA mutations (such as Figures 7F-7J). This could further clarify that the effects of the drug DAA or the impacts caused by RNA sequence mutations are indeed related to methylation modifications. In these KO cells, the effects caused by the drug or mutations are expected to diminish or disappear. These data would provide necessary support for the paper's conclusion.

5) The authors preliminarily verified the potential biological significance of G/M RNA methylation on viral infection in vivo using recombinant VSV containing SFTSV G. However, it is unclear whether the modification of G RNA within the VSV genomic circumstance in the context of VSV infection would occur and function in the same manner as during SFTSV infection. Further analyses of the rVSV infections are needed if the authentic challenge with SFTSV cannot be conducted.

**Part III – Minor Issues: Editorial and Data Presentation Modifications**

Reviewer #1: 1. The authors should confirm the changes of m6A levels in the cellular or viral genomes when the m6A writers or erasers are depleted or overexpressed.

2. Fig 6M, the difference in the NP level between WT and mutant viruses is moderate, when compared to viral RNA and titers in 6K/L. The authors may repeat or remove it.

3. Although the manuscript is professionally written, it has some grammatical errors and inappropriate wordings. For example, Line 129, is “precipitating” correct in its context? Do the authors mean “ participating” instead? Please proof-read the whole manuscript carefully.

Reviewer #2: The English writing is generally good, but some portions should be rewritten or revised to make it flows better and easier understood.

1. The authors put a lot of information that should be in the main text to help the readers to understand the experimental design into figure legends. For example, the in vivo experiment in mice, a brief description of the experimental design is lacking (Lines 587-589). The readers will have to get this information by digging into the long figure legends.

2. Line 139, "Current" makes the sentence confusing. It is not clear if the authors referred to their own manuscript or to the reference 41.

3. "RNA's" modification or "m6A's ability" have been used in few places. It will read better as modification of the viral RNA, as such.

Reviewer #3: 1）Fig S1: The methylation modification patterns observed in different cells vary significantly. Please provide further analysis or discussion.

2）Fig 3: The changes in protein expression are minimal and inconsistent. Overall, this result is relatively weak and could be considered for further analysis or placed in supplementary materials.

3）Fig 6: Are the predicted methylation patterns consistent with those actually identified? Why not refer to the results obtained from sequencing identification? How were the viruses packaged using reverse genetics amplified to high titers, and were the mutants sequenced for verification? Please provide more details necessary here.

4）In this study, how were the mRNAs of viral structural proteins and antigenome RNAs, as well as the NSs mRNA and S genome RNA, distinguished?

5) Fig 8A and 8B: The colors used for the two proteins are too similar; please use more contrasting colors to label different proteins.

6) line 34-35, please specify the use of surrogate viruses, rVSV.

7) line 95 and the following, METTL3 were written as METTTL3.

PLOS authors have the option to publish the peer review history of their article (what does this mean?). If published, this will include your full peer review and any attached files.

Reviewer #1: No

Reviewer #2: **Yes: **Fengwei Bai

Reviewer #3: No
---

## [Decision Letter · Decision Letter 1]

4 Nov 2024

Dear Dr. Dai,

We are pleased to inform you that your manuscript 'N6-methyladenosine RNA modification promotes Severe Fever with Thrombocytopenia Syndrome Virus infection' has been provisionally accepted for publication in PLOS Pathogens.

Best regards,

Holly Ramage

Guest Editor

PLOS Pathogens

Matthias Schnell

Section Editor

PLOS Pathogens

Michael Malim

Editor-in-Chief

PLOS Pathogens

orcid.org/0000-0002-7699-2064

Reviewer Comments (if any, and for reference):

Reviewer's Responses to Questions

**Part I - Summary**

Reviewer #1: The authors have addressed my previous critiques fully and satisfactorily. They have also fully addressed the other reviewers' comments with extensive new data.

Reviewer #2: The authors have addressed all my concerns.

Reviewer #3: (No Response)

**Part II – Major Issues: Key Experiments Required for Acceptance**

Reviewer #1: N/A

Reviewer #2: None

**Part III – Minor Issues: Editorial and Data Presentation Modifications**

Reviewer #1: N/A

Reviewer #2: None

Reviewer #3: (No Response)

PLOS authors have the option to publish the peer review history of their article (what does this mean?). If published, this will include your full peer review and any attached files.

Reviewer #1: No

Reviewer #2: No

Reviewer #3: No

---

## [Editor Report · Acceptance letter]

12 Nov 2024

Dear Dr. Dai,

We are delighted to inform you that your manuscript, "N6-methyladenosine RNA modification promotes Severe Fever with Thrombocytopenia Syndrome Virus infection," has been formally accepted for publication in PLOS Pathogens.

Best regards,

Michael Malim

Editor-in-Chief

PLOS Pathogens

orcid.org/0000-0002-7699-2064